# Verification and search algorithms for causal DAGs

**Davin Choo**[*]
National University of Singapore
`davin@u.nus.edu`

**Kirankumar Shiragur**[*]
Stanford University
`shiragur@stanford.edu`

**Arnab Bhattacharyya**
National University of Singapore
`arnabb@nus.edu.sg`

## Abstract

We study two problems related to recovering causal graphs from interventional data: (i) *verification*, where the task is to check if a purported causal graph is correct, and (ii) *search*, where the task is to recover the correct causal graph. For both, we wish to minimize the number of interventions performed. For the first problem, we give a characterization of a minimal sized set of atomic interventions that is necessary and sufficient to check the correctness of a claimed causal graph. Our characterization uses the notion of *covered edges*, which enables us to obtain simple proofs and also easily reason about earlier known results. We also generalize our results to the settings of bounded size interventions and node-dependent interventional costs. For all the above settings, we provide the first known provable algorithms for efficiently computing (near)-optimal verifying sets on general graphs. For the second problem, we give a simple adaptive algorithm based on graph separators that produces an atomic intervention set which fully orients any essential graph while using $\mathcal{O}(\log n)$ times the optimal number of interventions needed to *verify* (verifying size) the underlying DAG on $n$ vertices. This approximation is tight as *any* search algorithm on an essential line graph has worst case approximation ratio of $\Omega(\log n)$ with respect to the verifying size. With bounded size interventions, each of size $\leq k$, our algorithm gives an $\mathcal{O}(\log n \cdot \log k)$ factor approximation. Our result is the first known algorithm that gives a non-trivial approximation guarantee to the verifying size on general unweighted graphs and with bounded size interventions.

## 1 Introduction

Causal inference has long been an important concept in various fields such as philosophy [Rei56, Woo05, ES07], medicine/biology/genetics [KWJ+04, SC17, RHT+17, POS+18], and econometrics [Hoo90, RW06]. Recently, there has also been a growing interest in the machine learning community to use causal inference techniques to improve generalizability to novel testing environments (e.g. see [GUA+16, LKC17, ABGLP19, Sch22] and references therein). Under the assumption of causal sufficiency, where there is no unobserved confounders or selection bias, causal inference using observational data has been extensively studied and many algorithms such as PC [SGSH00] and GES [Chi02] have been proposed. These algorithms typically recover a causal graph up to its Markov equivalence class and it is known to be fundamentally impossible to learn causal relationships solely based on observational data. To overcome this issue, one either adds data modeling assumptions (e.g. see [SHHK06, PB14, MPJ+16]) or performs interventions to obtain interventional

---

[*]Equal contribution

36th Conference on Neural Information Processing Systems (NeurIPS 2022).

data (see Section 2.1 for a literature review). In our work, we study the causal discovery problem via interventions. As interventions often correspond to real-world experimental trials, they can be costly and it is of practical importance to minimize the number or cost of interventions.

In this work, we consider *ideal interventions* (i.e. hard interventions with infinite samples[2]) to recover causal graphs from its Markov equivalence class – the *Markov equivalence class* (MEC) of $G$, denoted by $[G]$, is the set of graphs that encode the same conditional distributions and it is known that any MEC $[G]$ can be represented by a unique partially oriented *essential graph* $\mathcal{E}(G)$. While ideal interventions may not always be possible in practice, they serve as a first step to understand the verification and search problems. Such interventions give us a clean graph-theoretic way to reason and identify arc directions in the causal graph $G = (V, E)$. With these interventions, it is known that intervening on a set $S \subseteq V$ allows us to infer the edge orientation of any edge cut by $S$ and $V \setminus S$ [Ebe07, HEH13, HLV14, SKDV15, KDV17].

Using the ideal interventions, we solve the verification and search problems. The *search problem* is the traditional question of finding a minimum set of interventions to recover the underlying ground truth causal graph. As for the *verification problem*, consider the following scenario: Suppose we have performed a observational study to obtain a MEC and consulted an expert about the identity of the ground truth causal graph. The question of verification involves testing if the expert is correct using the minimal number of interventional studies.

**Definition 1** (Search problem). Given the essential graph $\mathcal{E}(G^*)$ of an unknown causal graph $G^*$, use the minimal number of interventions to fully recover the ground truth causal graph $G^*$.

**Definition 2** (Verification problem). Given the essential graph $\mathcal{E}(G^*)$ of an unknown causal graph $G^*$ and an expert's graph $G \in [G^*]$, use the minimal number of interventions to verify $G \stackrel{?}{=} G^*$.

To solve both these problems, we compute *verifying sets* (see Definition 4). A verifying set is a collection of interventions that fully orients the essential graph. Note that the minimum size/cost verifying set serves as a natural lower bound for both the search and verification problems. One of our key contributions is to efficiently compute these verifying sets for a given causal graph.

**Contributions** We study the problems of verification and search for a causal graph from its MEC using ideal interventions. In our work, we make standard assumptions such as the Markov assumption, the faithfulness assumption, and causal sufficiency [SGSH00].

1. **Verification**: We provide the first known efficient algorithms for computing minimal sized atomic verifying sets and near-optimal bounded size verifying sets (that use at most one more intervention than optimal) on general graphs. When vertices have additive interventional costs according to a weight function $w : V \to \mathbb{R}$, we give efficient computation of verifying sets $\mathcal{I}$ that minimizes $\alpha \cdot w(\mathcal{I}) + \beta \cdot |\mathcal{I}|$. Our atomic verifying sets have optimal cost and bounded size verifying sets incur a total additive cost of at most $2\beta$ more than optimal. To achieve these results, we prove properties about covered edges and give a characterization of verifying sets as a separation of unoriented covered edges in the given essential graph. Using our covered edge perspective, we show that the universal lower bounds of [SMG+20, PSS22] are *not* tight and give a simple proof recovering the verification upper bound of [PSS22].

2. **Adaptive search**: We consider adaptive search algorithms which produce a *sequence* of interventions one-at-a-time, possibly using information gained from the outcomes of earlier chosen interventions. Building upon the lower bound of [SMG+20], we establish a stronger (but not computable) lower bound to verify a causal graph. This further implies a lower bound on the minimum interventions needed by *any* adaptive search algorithm. We also provide an adaptive search algorithm (based on graph separators for chordal graphs) and use our lower bound to prove that our approach uses at most a logarithmic multiplicative factor more interventions than a minimum sized verifying set of the true underlying causal graph.

**Outline** Section 2 introduces notation and preliminary notions, as well as discuss some known results on ideal interventions. We give our results in Section 3 and provide an overview of the techniques used in Section 4. We discuss our experimental results in Section 5. Full proofs, side discussions, and source code/scripts are given in the appendix.

---

[2]We do not consider sample complexity issues in this work.

## 2 Preliminaries and related work

For any set $A$, we denote its powerset by $2^A$. We write $\{1, \ldots, n\}$ as $[n]$ and hide absolute constant multiplicative factors in $n$ using asymptotic notations $\mathcal{O}(\cdot)$, $\Omega(\cdot)$, and $\Theta(\cdot)$. Throughout, we use $G^*$ to denote the (unknown) ground truth DAG and we only know its essential graph $\mathcal{E}(G^*)$.

### Graph notions

We study partially directed graphs $G = (V, E, A)$ on $|V| = n$ vertices with unoriented edges $E$ and oriented arcs $A$. Any possible edge between two distinct vertices $u, v \in V$ is either undirected, oriented in one direction, or absent in the graph. We write $u \sim v$ if these vertices are connected (either through an unoriented edge or an arc) in the graph and $u \nsim v$ to indicate the absence of any edge/arc connection. If $(u, v) \in A$ or $(v, u) \in A$, we write $u \to v$ or $u \leftarrow v$ respectively. When $E = \emptyset$, we say that the graph is fully oriented. For any graph $G$, we use $V(G), E(G), A(G)$ to denote its vertices, unoriented edges and oriented arcs.

For any subset of vertices $V' \subseteq V$, $G[V']$ denotes the vertex-induced subgraph on $V'$. Similarly, we define $G[A']$ and $G[E']$ as arc-induced and edge-induced subgraphs of $G$ for $A' \subseteq A$ and $E' \subseteq E$ respectively. For an undirected graph $G$, $\omega(G)$ refers to the size of its maximum clique and $\chi(G)$ refers to its chromatic number. For any vertex $v \in V$ in a directed graph, $Pa(v) \subseteq V$ denotes the parent set of $v$ and $pa(v)$ denotes the vector of values taken by $v$'s parents.

The *skeleton* of a graph $G$ refers to the graph $G' = (V, E \cup A, \emptyset)$ where all arcs are made unoriented. A *v-structure* refers to three distinct vertices $u, v, w \in V$ such that $u \to v \leftarrow w$ and $u \nsim w$. A simple cycle is a sequence of $k \geq 3$ vertices where $v_1 \sim v_2 \sim \ldots \sim v_k \sim v_1$. The cycle is directed if at least one of the edges is directed and all directed arcs are in the same direction along the cycle. A partially directed graph is a *chain graph* if it contains no directed cycle. In the undirected graph $G'$ obtained by removing all arcs from a chain graph $G$, each connected component in $G'$ is called a *chain component*. We use $CC(G)$ to denote the set of chain components in $G$. Note that the vertices of these chain components form a partition of $V$.

Directed acyclic graphs (DAGs), a special case of chain graphs where *all* edges are directed, are commonly used as graphical causal models [Pea09] where vertices represents random variables and the joint probability density $f$ factorizes according to the Markov property: $f(v_1, \ldots, v_n) = \prod_{i=1}^n f(v_i \mid pa(v))$. We can associate a *valid permutation / topological ordering* $\pi : V \to [n]$ to any (partially oriented) DAG such that oriented arcs $(u, v)$ satisfy $\pi(u) < \pi(v)$ and any unoriented arc $\{u, v\}$ can be oriented as $u \to v$ whenever $\pi(u) < \pi(v)$ without forming directed cycles. While there may be multiple valid permutations, we often only care that there exists at least one such permutation. For any DAG $G$, we denote its *Markov equivalence class* (MEC) by $[G]$ and *essential graph* by $\mathcal{E}(G)$. It is known that two graphs are Markov equivalent if and only if they have the same skeleton and v-structures [VP90, AMP97].

A *clique* is a graph where $u \sim v$ for any pair of vertices $u, v \in V$. A *maximal clique* is an vertex-induced subgraph of a graph that is a clique and ceases to be one if we add any other vertex to the subgraph. If all edges in the clique are oriented in an acyclic manner, then there is a unique valid permutation $\pi$ that respects this orientation. We denote $v = \text{argmax}_{u \in V} \pi(u)$ as the *sink* of the clique.

### Interventions, verifying sets, and additive vertex intervention costs

An *intervention* $S \subseteq V$ is an experiment where the experimenter forcefully sets each variable $s \in S$ to some value, independent of the underlying causal structure. An intervention is called an *atomic intervention* if $|S| = 1$ and called a *bounded size intervention* if $|S| \leq k$ for some size upper bound $k$. One can view observational data as a special case where $S = \emptyset$. Interventions affect the joint distribution of the variables and are formally captured by Pearl's do-calulus [Pea09]. An *intervention set* $\mathcal{I} \subseteq 2^V$ is a collection of interventions and $\cup_{S \in \mathcal{I}} S$ is the union of all intervened vertices.

In this work, we study ideal interventions. Graphically speaking, an ideal intervention $S$ on $G$ induces an interventional graph $G_S$ where all incoming arcs to vertices $v \in S$ are removed [EGS12] and it is known that intervening on a set $S \subseteq V$ allows us to infer the edge orientation of any edge cut by $S$ and $V \setminus S$ [Ebe07, HEH13, HLV14, SKDV15, KDV17]. For ideal interventions, an $\mathcal{I}$-essential graph $\mathcal{E}_{\mathcal{I}}(G)$ of $G$ is the essential graph representing the Markov equivalence class of graphs whose interventional graphs for each intervention is Markov equivalent to $G_S$ for any intervention $S \in \mathcal{I}$.

There are several known properties about $\mathcal{I}$-essential graph properties [HB12, HB14, SMG$^+$20]: Every $\mathcal{I}$-essential graph is a chain graph with chordal[3] chain components. This includes the case of $S = \emptyset$. Orientations in one chain component do not affect orientations in other components. In other words, to fully orient any essential graph $\mathcal{E}(G^*)$, it is necessary and sufficient to orient every chain component in $\mathcal{E}(G^*)$ independently. More formally, we have[4]

**Lemma 3** (Modified lemma 1 of [HB14]). *Let $\mathcal{I} \subseteq 2^V$ be an intervention set. Consider the $\mathcal{I}$-essential graph $\mathcal{E}_{\mathcal{I}}(G^*)$ of some DAG $G^*$ and let $H \in CC(\mathcal{E}_{\mathcal{I}}(G^*))$ be one of its chain components. Then, for any additional interventional set $\mathcal{I}' \subseteq 2^V$ such that $\mathcal{I} \cap \mathcal{I}' = \emptyset$, we have*

$$\mathcal{E}_{\mathcal{I} \cup \mathcal{I}'}(G^*)[V(H)] = \mathcal{E}_{\{S \cap V(H) \,:\, S \in \mathcal{I}'\}}(G^*[V(H)]).$$

As a consequence of Lemma 3, one may assume without loss of generality that $CC(\mathcal{E}(G))$ is a single connected component and then generalize results by summing across all connected components.

A *verifying set* $\mathcal{I}$ for a DAG $G \in [G^*]$ is an intervention set that fully orients $G$ from $\mathcal{E}(G^*)$, possibly with repeated applications of Meek rules (see Appendix A). In other words, for any graph $G = (V, E)$ and any verifying set $\mathcal{I}$ of $G$, we have $\mathcal{E}_{\mathcal{I}}(G)[V'] = G[V']$ for *any* subset of vertices $V' \subseteq V$. Furthermore, if $\mathcal{I}$ is a verifying set for $G$, then $\mathcal{I} \cup S$ is also a verifying set for $G$ for any additional intervention $S \subseteq V$. While DAGs may have multiple verifying sets in general, we are often interested in finding one with minimum size or cost.

**Definition 4** (Minimum size/cost verifying set). Let $w$ be a weight function on intervention sets. An intervention set $\mathcal{I}$ is called a verifying set for a DAG $G^*$ if $\mathcal{E}_{\mathcal{I}}(G^*) = G^*$. $\mathcal{I}$ is a *minimum size (resp. cost) verifying set* if $\mathcal{E}_{\mathcal{I}'}(G^*) \neq G^*$ for any $|\mathcal{I}'| < |\mathcal{I}|$ (resp. for any $w(\mathcal{I}') < w(\mathcal{I})$).

When restricting to interventions of size at most $k$, the *minimum verification number* $\nu_k(G)$ of $G$ denotes the size of the minimum size verifying set for any DAG $G \in [G^*]$. That is, any revealed arc directions when performing interventions on $\mathcal{E}(G^*)$ respects $G$. $\nu_1(G)$ denotes the case where we restrict to atomic interventions. One of the goals of this work is to characterize $\nu(G)$ given an essential graph $\mathcal{E}(G^*)$ and some $G \in [G^*]$.

**Covered edges**

Covered edges are special arcs in a causal graph where the endpoints of $u \sim v$ share the same set of parents in $V \setminus \{u, v\}$. These edges are crucial in causal discovery because their orientation can be reversed and they still yield the same conditional independencies. See Fig. 1 for an illustration. Note that one can compute all covered edges of a given DAG $G$ in polynomial time.

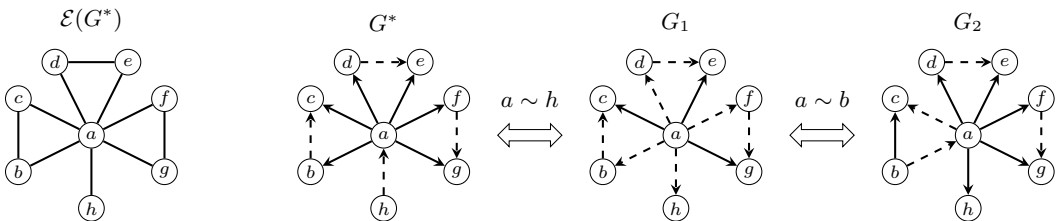

Figure 1: A DAG $G^*$ with its essential graph $\mathcal{E}(G^*)$ on the left. $G_1$ and $G_2$ are two other DAGs that belong to the same Markov equivalence class $[G^*]$. Dashed arcs are covered edges in each DAG. One can perform a sequence of covered edge reversals to transform between the DAGs (see Lemma 7). Note that the sizes of the minimum vertex cover of the covered edges may differ across DAGs.

**Definition 5** (Covered edge). An edge $u \sim v$ is a covered edge if $Pa(u) \setminus \{v\} = Pa(v) \setminus \{u\}$.

**Definition 6** (Covered edge reversal). A covered edge reversal means that we replace $u \to v$ with $v \to u$, for some covered edge $u \to v$, while keeping all other arcs unchanged.

---

[3]A chordal graph is a graph where every cycle of length at least 4 has a chord, which is an edge that is not part of the cycle but connects two vertices of the cycle. See [BP93] for more properties.

[4]Lemma 1 of [HB14] actually considers a *single* additional intervention, but a closer look at their proof shows that the statement can be strengthened to allow for *multiple* additional interventions. For completeness, we provide the proof of this strengthened version in Appendix B. Note that we can drop the $\emptyset$ intervention in the statement since essential graphs are defined with the observational data provided.

**Lemma 7** ([Chi95])**.** *If $G$ and $G'$ belong in the same MEC if and only if there exists a sequence of covered edge reversals to transform between them.*

**Definition 8** (Separation of covered edges)**.** We say that an intervention $S \subseteq V$ *separates* a covered edge $u \sim v$ if $|\{u, v\} \cap S| = 1$. That is, *exactly* one of the endpoints is intervened by $S$. We say that an intervention set $\mathcal{I}$ separates a covered edge $u \sim v$ if there exists $S \in \mathcal{I}$ that separates $u \sim v$.

**Relationship between searching and verification**

Recall Definition 1 and Definition 2. The verification number is a useful analytical tool for the search problem as $\nu_k(G^*)$ is a lower bound on the number of interventions used by an optimal search algorithm. Furthermore, since the search problem needs to fully orient $\mathcal{E}(G^*)$ regardless of which DAG is the ground truth, any search algorithm given $\mathcal{E}(G^*)$ requires *at least* $\min_{G \in [G^*]} \nu_k(G)$ interventions, even if it is adaptive and randomized. In fact, the *strongest possible universal lower bound* guarantee one can prove must be *at most* $\min_{G \in [G^*]} \nu_k(G)$ and the *strongest possible universal upper bound* guarantee one can prove must be *at least* $\max_{G \in [G^*]} \nu_k(G)$. Note that if the search algorithm is *non-adaptive*, then it trivially needs *at least* $\max_{G \in [G^*]} \nu_k(G)$ interventions.

**Example** Consider the graph $G^*$ in Fig. 1 where the essential graph $\mathcal{E}(G^*)$ representing the MEC $[G^*]$ is the standing windmill[5]. Now, consider only atomic interventions. We will later show that the minimum verification number of a DAG is the size of the minimum vertex cover of its covered edges (see Theorem 11). One can check that $\nu_1(G^*) = \nu_1(G_1) = 4$ while $\nu_1(G_2) = 3$. In fact, we actually show that $\min_{G \in [G^*]} \nu_1(G) = 3$ and $\max_{G \in [G^*]} \nu_1(G) = 4$ in Appendix C. Thus, any search algorithm using only atomic interventions on $\mathcal{E}(G^*)$ needs at least 3 atomic interventions.

## 2.1 Related work

Table 1 and Table 2 in Appendix D summarize[6] the existing upper (sufficient) and lower (worst case necessary) bounds on the size ($|\mathcal{I}|$, or $\mathbb{E}[|\mathcal{I}|]$ for randomized algorithms) of intervention sets that fully orient a given essential graph. These lower bounds are "worst case" in the sense that there exists a graph, typically a clique, which requires the stated number of interventions. Observe that there are settings where adaptivity[7] and randomization strictly improves the number of required interventions.

**Separating systems** [HEH13] drew connections between causal discovery via interventions and the concept of separating systems from the combinatorics literature. This was extended by [SKDV15] to the bounded size and adaptive settings. An $(n, k)$-separating system is a Boolean matrix with $n$ columns where each row has at most $k$ ones, indicating which vertex is to be intervened upon. Using their proposed separating system construction based on "label indexing", [SKDV15] showed that roughly $\frac{n}{k} \log_{\frac{n}{k}} n$ interventions is sufficient to fully an essential graph $G$ with bounded size interventions. On cliques (i.e. worst case lower bound), [SKDV15] showed that the bound is tight while only roughly $\frac{\chi(\mathcal{E}(G))}{k} \log_{\frac{\chi(\mathcal{E}(G))}{k}} \chi(\mathcal{E}(G))$ interventions are necessary for general graphs[8], even if the interventions are chosen adaptively or in a randomized fashion.

**Universal bounds for minimum sized atomic interventions** Beyond worst case lower bounds, recent works have studied universal bounds for orienting essential graphs $\mathcal{E}(G^*)$ using atomic interventions [SMG$^+$20, PSS22]. These universal bounds depend on graph parameters of $\mathcal{E}(G)$ beyond the number of nodes $n$. [SMG$^+$20] showed that search algorithms must use at least $\sum_{H \in CC(\mathcal{E}(G^*))} \lfloor \frac{\omega(H)}{2} \rfloor$ interventions, where $H$ is a chain component of $\mathcal{E}(G^*)$ and the summation across chain components is a consequence of Lemma 3. They also introduced a graph concept called directed clique trees and designed an adaptive, deterministic algorithm. On intersection-incomparable chordal graphs, their algorithm outputs an intervention set of size $\mathcal{O}(\log_2(\max_{H \in CC(\mathcal{E}(G^*))} \omega(H)) \cdot \nu_1(G^*))$. More recently, [PSS22] introduced the notion of clique-block shared-parents orderings and showed that

---

[5]To be precise, it is the Wd(3,3) windmill graph with an additional edge from the center.

[6]Some known results are discussed in further detail below instead of being summarized in the table format.

[7]Given an essential graph $\mathcal{E}(G^*)$, non-adaptive algorithms decide a *set* of interventions without looking at the outcomes of the interventions. Meanwhile, adaptive algorithms can provide a *sequence* of interventions one-at-a-time, possibly using any information gained from the outcomes of earlier chosen interventions.

[8]Note that there is a slight gap between $\frac{\chi(\mathcal{E}(G))}{k} \log_{\frac{\chi(\mathcal{E}(G))}{k}} \chi(\mathcal{E}(G))$ and $\frac{n}{k} \log_{\frac{n}{k}} n$ on general graphs.

any search algorithm for an essential graph $\mathcal{E}(G^*)$ with $r$ maximal cliques requires at least $\lceil \frac{n-r}{2} \rceil$ interventions and $\nu_1(G) \leq n - r$ for any $G \in [G^*]$.

**Non-atomic interventions** The randomized algorithm of [HLV14] fully orients an essential graph using $\mathcal{O}(\log(\log(n)))$ unbounded interventions in expectation. Building upon this, [SKDV15] shows that $\mathcal{O}(\frac{n}{k}\log(\log(k)))$ bounded sized interventions (each involving at most $k$ nodes) suffice.

**Additive vertex costs** [KDV17, GSKB18, LKDV18] studied the *non-adaptive* search setting where vertices may have different intervention costs and intervention costs accumulate additively. [GSKB18] studied the problem of maximizing number of oriented edges given a budget of atomic interventions while [KDV17, LKDV18] studied the problem of finding a minimum cost (bounded size) intervention set that fully orients the essential graph. [LKDV18] showed that computing the minimum cost intervention set is NP-hard and gave search algorithms with constant approximation factors.

**Other related work** [HLV14, KSSU19] showed that Erdős-Rényi graphs can be easily oriented.

## 3 Results

### 3.1 Verification

Our core contribution for the verification problem is deriving an interesting connection between the covered edges and verifying sets. We show importance of this connection by using it to derive several novel results on finding optimal verifying sets in various settings such as bounded size interventions and when vertices have varying interventional costs. For detailed proofs, see Appendix E.

**Theorem 9.** *Fix an essential graph $\mathcal{E}(G^*)$ and $G \in [G^*]$. An intervention set $\mathcal{I}$ is a verifying set for $G$ if and only if $\mathcal{I}$ is a set that separates every covered edge of $G$ that is unoriented in $\mathcal{E}(G^*)$.*

Together with Lemma 7 (any undirected edge in $\mathcal{E}(G^*)$ is a covered edge for *some* $G \in [G^*]$), Theorem 9 implies a simple alternative proof for an earlier known result that characterizes non-adaptive search algorithms via separating systems [HEH13, SKDV15]: any non-adaptive search algorithm, which has *no* knowledge of $G^*$, should separate *every* undirected edge in $\mathcal{E}(G^*)$.

Another immediate application of Theorem 9 is the following result for the verification problem.

**Corollary 10.** *Given an essential graph $\mathcal{E}(G^*)$ of an unknown ground truth DAG $G^*$ and a causal DAG $G \in [G^*]$, we can test if $G \stackrel{?}{=} G^*$ by intervening on any verifying set of $G$. Furthermore, in the worst case, any algorithm that correctly resolves $G \stackrel{?}{=} G^*$ needs at least $\nu(G)$ interventions.*

The above corollary provides a solution to the verification problem in terms of verifying sets and in the following we give efficient algorithms for computing these verifying sets that are optimal in the atomic setting and are near-optimal in the case of bounded size.

**Theorem 11.** *Fix an essential graph $\mathcal{E}(G^*)$ and $G \in [G^*]$. An atomic intervention set $\mathcal{I}$ is a minimal sized verifying set for $G$ if and only if $\mathcal{I}$ is a minimum vertex cover of unoriented covered edges of $G$. A minimal sized atomic verifying set can be computed in polynomial time.*

Our result provides the first efficient algorithm for computing minimum sized atomic verifying set for general graphs. Previously, efficient algorithms for computing minimum sized atomic verifying sets were only known for simple graphs such as cliques and trees. For general graphs, only a brute force algorithm is known [SMG$^+$20, Appendix F] which takes exponential time in the worst case[9]. In contrast to optimal algorithms, [PSS22] provides an efficient algorithm that returns a verifying set of size at most 2 times that of the optimum.

**Theorem 12.** *Fix an essential graph $\mathcal{E}(G^*)$ and $G \in [G^*]$. If $\nu_1(G) = \ell$, then $\nu_k(G) \geq \lceil \frac{\ell}{k} \rceil$ and there exists a polynomial time algo. to compute a bounded size intervention set $\mathcal{I}$ of size $|\mathcal{I}| \leq \lceil \frac{\ell}{k} \rceil + 1$.*

To the best of our knowledge, our work provides the first known efficient algorithm for computing near-optimal bounded sized verifying sets for general graphs. Furthermore, we note that, for every $k$,

---

[9]Theorem 11 also provides a rigorous justification to the observation of [SMG$^+$20] that "In general, the size of an [atomic verifying set] cannot be calculated from just its essential graph". This is because essential graphs could imply minimum vertex covers of different sizes (see Fig. 1).

there exists a family of graphs where the optimum solution requires at least $\lceil \frac{\ell}{k} \rceil + 1$ bounded size interventions. Thus, our upper bound is tight in the worst case (see Fig. 6 in Appendix E.5).

Beyond minimal sized interventions, a natural and much broader setting in causal inference is one where different vertices have varying intervention costs, e.g. in a smoking study, it is easier to modify a subject's diet than to force the subject to smoke (or stop smoking). Formally, one can define a weight function on the vertices $w : V \to \mathbb{R}$ which overloads to $w(S) = \sum_{v \in S} w(v)$ on interventions and $w(\mathcal{I}) = \sum_{S \in \mathcal{I}} S$ on intervention sets. Such an additive cost structure has been studied by [KDV17, GSKB18]. Consider an essential graph which is a star graph on $n$ nodes where the leaves have cost 1 and the root has cost significantly larger than $n$. For atomic verifying sets, we see that the *minimum cost* verifying set is to intervene on the leaves while the *minimum size* verifying set is to simply intervene on the root. Since one may be more preferred over the other, depending on the actual real-life situation, we propose to find a verifying set $\mathcal{I}$ which minimizes

$$\alpha \cdot w(\mathcal{I}) + \beta \cdot |\mathcal{I}| \qquad \text{where } \alpha, \beta \geq 0 \tag{1}$$

so as to explicitly trade-off between the cost and size of the intervention set. This objective also naturally allows the constraint of bounded size interventions by restricting $|S| \leq k$ for all $S \in \mathcal{I}$. Note that the earlier results on minimum size verifying sets correspond to $\alpha = 0$ and $\beta = 1$. Interestingly, the techniques we developed in the minimum size setting generalizes to this broader setting. Furthermore, our framework of covered edges allow us to get simple proofs.

**Theorem 13.** *Fix an essential graph $\mathcal{E}(G^*)$ and $G \in [G^*]$. An atomic verifying set $\mathcal{I}$ for $G$ that minimizes Eq. (1) can be computed in polynomial time.*

**Theorem 14.** *Fix an essential graph $\mathcal{E}(G^*)$ and $G \in [G^*]$. Suppose the optimal bounded size intervention set that minimizes Eq. (1) costs $OPT$. Then, there exists a polynomial time algorithm that computes a bounded size intervention set with total cost $OPT + 2\beta$.*

## 3.2 Adaptive search

Here, we study the unweighted search problem in causal inference where one wishes to fully orient an essential graph obtained from observational data while minimizing the number of interventions used. Formally, we give an algorithm (Algorithm 1) that fully orients $\mathcal{E}(G^*)$ using at most a logarithmic multiplicative factor more interventions than $\nu_k(G^*)$, the number of (bounded size) interventions needed to *verify* the ground truth $G^*$. Since *any* search algorithm will incur at least $\nu_k(G^*)$ interventions, our result implies that search is (almost, up to $\log n$ multiplicative approximation) as easy as the verification. For detailed proofs, see Appendix F.

**Theorem 15.** *Fix an essential graph $\mathcal{E}(G^*)$ with an unknown underlying ground truth DAG $G^*$. Given $k = 1$, Algorithm 1 runs in polynomial time and computes an atomic intervention set $\mathcal{I}$ in a deterministic and adaptive manner such that $\mathcal{E}_{\mathcal{I}}(G^*) = G^*$ and $|\mathcal{I}| \in \mathcal{O}(\log(n) \cdot \nu_1(G^*))$.*

**Theorem 16.** *Fix an essential graph $\mathcal{E}(G^*)$ with an unknown underlying ground truth DAG $G^*$. Given $k > 1$, Algorithm 1 runs in polynomial time and computes a bounded size intervention set $\mathcal{I}$ in a deterministic and adaptive manner such that $\mathcal{E}_{\mathcal{I}}(G^*) = G^*$ and $|\mathcal{I}| \in \mathcal{O}(\log(n) \cdot \log(k) \cdot \nu_k(G^*))$.*

These results are the first competitive results that holds for using atomic or bounded size interventions on *general graphs*. The only previously known result of $\mathcal{O}(\log_2(\max_{H \in CC(\mathcal{E}(G^*))} \omega(H)) \cdot \nu_1(G^*))$ by [SMG+20] was an algorithm based on directed clique trees with provable guarantees only for atomic interventions on intersection-incomparable chordal graphs. To obtain our results, we are *not* simply improving the analysis of [SMG+20]. Instead, we developed a new algorithmic approach that is based on graph separators which is a much simpler concept than directed clique trees.

The approximation of $\mathcal{O}(\log n)$ to $\nu_1(G^*)$ is the tightest one can hope for atomic interventions in general. For instance, consider the case where $\mathcal{E}(G^*)$ is an undirected line graph on $n$ vertices. Then, *any* adaptive algorithm needs $\Omega(\log n)$ atomic interventions in the worst case[10] while $\nu_1(G^*) = 1$. The line graph also provides a clear distinction between adaptive and non-adaptive search algorithms since *any* non-adaptive algorithm needs $\Omega(n)$ atomic interventions to separate all the edges in $\mathcal{E}(G^*)$.

---

[10]The lower bound reasoning is similar to the lower bound for binary search.

# 4 Overview of techniques

## 4.1 Verification

Our results for the verification problem are broadly divided into two categories: (I) Connection between the covered edges and verification set (Theorem 9, Corollary 10); (II) Efficient computation of verification sets under various settings using the connection established.

For the first type of results, we show that any intervention set is a verifying set if and only if it separates every unoriented covered edge of $G \in \mathcal{E}(G^*)$. For necessity, we show that all four Meek rules (which are known to be consistent and complete) will *not* orient any unoriented covered edge of $G$ that is *not* separated by any intervention. Our proof is simple due to the usage of covered edges. For sufficiency, we show that *every* unoriented non-covered edge of $G$ will be oriented by Meek rules if all covered edges are separated. We prove this using a subtle induction over a valid topological ordering of the vertices $\pi$ of $G^*$: Let $V_i$ be the first $i$ smallest vertices in $\pi$, for $i = 1, 2, \ldots, n$. Consider subgraph $\mathcal{E}(G^*)[V_i]$ induced by $V_i$ with $v_i$ being the last vertex in the ordering of $V_i$. By induction, it suffices to show that all non-covered $u \to v_i$ edges are oriented for $u \in V_{i-1}$. To show this, we perform case analysis to argue that either $u \to v_i$ is part of v-structure (i.e. is already oriented in $\mathcal{E}(G^*)$) or Meek rule R2 will orient it.

The previous result only establishes equivalence between the verifying sets and the intervention sets that separate unoriented covered edges. For efficient computation of optimal verifying sets, we prove several additional properties of covered edges, which may be of independent interest.

**Lemma 17** (Properties of covered edges)**.**

1. *Let $H$ be the edge-subgraph induced by covered edges of a DAG $G$. Then, every vertex in $H$ has at most one incoming edge and thus $H$ is a forest of directed trees.*

2. *If a DAG $G$ is a clique on $n \geq 3$ vertices $v_1, v_2, \ldots, v_n$ with $\pi(v_1) < \pi(v_2) < \ldots < \pi(v_n)$, then $v_1 \to v_2, \ldots, v_{n-1} \to v_n$ are the covered edges of $G$.*

3. *If $u \to v$ is a covered edge in a DAG $G$, then $u$ cannot be a sink of any maximal clique of $G$.*

The fact that the edge-induced subgraph of the covered edges is a forest enables us to use standard dynamic programming techniques to compute (weighted) minimum vertex covers for the unoriented covered edges of $G$, corresponding to minimum size atomic verifying sets (Theorem 11 and Theorem 13). For bounded size verifying sets, we exploit the fact that trees are bipartite and so we can divide the minimum vertex covers into two partitions. Since vertices within each partite are non-adjacent, we can group them into larger interventions without affecting the overall number of separated edges, giving us the guarantees in Theorem 12 and Theorem 14.

Through the lens of covered edges, we see that existing universal bounds of [SMG$^+$20, PSS22] are *not* tight[11]. Consider the case where the essential graph $\mathcal{E}(G^*)$ is the standing windmill graph given in Fig. 1. The graph $\mathcal{E}(G^*)$ has $n = 8$ nodes, $r = 4$ maximal cliques and the largest maximal clique is size 3. The lower bound of [SMG$^+$20] yields $\sum_{H \in CC(\mathcal{E}(G^*))} \lfloor \frac{\omega(H)}{2} \rfloor = \lfloor \frac{3}{2} \rfloor = 1$ while lower bound of [PSS22] yields $\lceil \frac{n-r}{2} \rceil = \lceil \frac{8-4}{2} \rceil = 2$. Meanwhile, we show $\min_{G \in [G^*]} \nu_1(G) = 3$ in Appendix C.

We can also recover the $\nu_1(G^*) \leq n - r$ bound of [PSS22] with a short proof using covered edges.

**Lemma 18.** *For any essential graph $\mathcal{E}(G^*)$ on $n$ vertices with $r$ maximal cliques, there exists an atomic verifying set of size at most $n - r$.*

*Proof.* By Theorem 11, it suffices to find a vertex cover of the unoriented covered edges of $\mathcal{E}(G^*)$. By Lemma 17, any covered edge $u \to v$ cannot have $u$ as a sink of any maximal clique. So, the set of all vertices without the $r$ sink vertices is a vertex cover of the covered edges in $\mathcal{E}(G^*)$. $\square$

## 4.2 Search

Here we provide the description and proof overview of the search results. Our search algorithm (Algorithm 1) relies on graph separators that we formally define next. Existence and efficient

---

[11]The non-tightness of the universal lower bound of [PSS22] was known and verified via random graph experiments.

computation of graph separators are well studied [LT79, GHT84, GRE84, AST90, KR10, WN11] and are commonly used in divide-and-conquer graph algorithms and as analysis tools.

**Definition 19** ($\alpha$-separator and $\alpha$-clique separator). *Let $A, B, C$ be a partition of the vertices $V$ of a graph $G = (V, E)$. We say that $C$ is an $\alpha$-separator if no edge joins a vertex in $A$ with a vertex in $B$ and $|A|, |B| \leq \alpha \cdot |V|$. We call $C$ is an $\alpha$-clique separator if it is an $\alpha$-separator and a clique.*

---

**Algorithm 1** Search algorithm via graph separators.

---

1: **Input**: Essential graph $\mathcal{E}(G^*)$, intervention size $k$. **Output**: A fully oriented graph $G \in [G^*]$.
2: Initialize $i = 0$ and $\mathcal{I}_0 = \emptyset$.
3: **while** $\mathcal{E}_{\mathcal{I}_i}(G^*)$ still has undirected edges **do**
4:     For each $H \in CC(\mathcal{E}_{\mathcal{I}_i}(G^*))$ of size $|H| \geq 2$, find a 1/2-clique separator $K_H$ using Theorem 20. Define $Q = \{K_H\}_{H \in CC(\mathcal{E}_{\mathcal{I}_i}(G^*)), |H| \geq 2}$ as the union of clique separator nodes.
5:     **if** $k = 1$ or $|Q| = 1$ **then** Define $C_i = Q$ as an atomic intervention set.
6:     **else** Define $k' = \min\{k, |Q|/2\}$, $a = \lceil |Q|/k' \rceil \geq 2$, and $\ell = \lceil \log_a n \rceil$. Compute labelling scheme of [SKDV15, Lemma 1] on $Q$ with $(|Q|, k', a)$, and define $C_i = \{S_{x,y}\}_{x \in [\ell], y \in [a]}$, where $S_{x,y} \subseteq Q$ is the subset of vertices whose $x^{th}$ letter in the label is $y$.
7:     Update $i \leftarrow i + 1$, intervene on $C_i$ to obtain $\mathcal{E}_{\mathcal{I}_i}(G^*)$, and update $\mathcal{I}_i \leftarrow \mathcal{I}_{i-1} \cup C_i$.
8: **end while**

---

We first give the proof strategy for Theorem 15, the atomic case where $k = 1$. We divide the analysis into two steps. In the first step, we show that the algorithm terminates in $\mathcal{O}(\log n)$ iterations. In the second step, we argue that the number of interventions performed in each iteration is in $\mathcal{O}(\nu_1(G))$.

The analysis of the first step relies on the following result of [GRE84].

**Theorem 20** ([GRE84], instantiated for unweighted graphs). *Let $G = (V, E)$ be a chordal graph with $|V| \geq 2$ and $p$ vertices in its largest clique. There exists a $1/2$-clique-separator $C$ of size $|C| \leq p - 1$. The clique $C$ can be computed in $\mathcal{O}(|E|)$ time.*

At each iteration $i$ of the algorithm, we intervene on a $1/2$-clique separators of each connected chain component. Note that the size of each connected chain compoenent at the end of iteration $i$ is at most $n/2^i$. Therefore, after $\mathcal{O}(\log n)$ iterations, each connected chain component will contain at most 1 vertex, implying that all the edges have been oriented.

To bound the number of interventions used in each iteration, we prove a stronger universal lower bound that is built upon the lower bound of [SMG⁺20]. While *not* computable[12], it is a very powerful lower bound for analysis: In Fig. 7 of Appendix G, we give an example where $\nu_1(G^*) \approx n$ while the lower bound of [SMG⁺20] on $CC(\mathcal{E}(G^*))$ is a constant. Meanwhile, there exists a set of atomic interventions $\mathcal{I}$ such that applying [SMG⁺20] on $CC(\mathcal{E}_{\mathcal{I}}(G^*))$ yields a much stronger $\Omega(n)$ bound.

**Lemma 21.** *Fix an essential graph $\mathcal{E}(G^*)$ and $G \in [G^*]$. Then,*

$$\nu_1(G) \geq \max_{\mathcal{I} \subseteq V} \sum_{H \in CC(\mathcal{E}_{\mathcal{I}}(G^*))} \left\lfloor \frac{\omega(H)}{2} \right\rfloor$$

As we intervene on cliques in each connected component at every iteration, Lemma 21 shows that we use at most $2 \cdot \nu_1(G^*)$ interventions per iteration. Therefore, the total interventions used by Algorithm 1 is in $\mathcal{O}(\log(n) \cdot \nu_1(G^*))$.

For bounded size interventions (Theorem 16), we follow the same strategy as above but we modify how we orient the edges within and cut by the $1/2$-clique separators. More specifically, we compute a separating system for the union of clique separator nodes based on the labeling scheme of [SKDV15] and perform case analysis to argue that we use $\mathcal{O}(\log(k) \cdot \nu_k(G^*))$ bounded size interventions per iteration.

**Lemma 22** (Lemma 1 of [SKDV15]). *Let $(n, k, a)$ be parameters where $k \leq n/2$. There is a polynomial time labeling scheme that produces distinct $\ell$ length labels for all elements in $[n]$ using letters from the integer alphabet $\{0\} \cup [a]$ where $\ell = \lceil \log_a n \rceil$. Further, in every digit (or position), any integer letter is used at most $\lceil n/a \rceil$ times. This labelling scheme is a separating system: for any $i, j \in [n]$, there exists some digit $d \in [\ell]$ where the labels of $i$ and $j$ differ.*

---

[12]It involves a maximization over all possible atomic interventions *and* we do not know the $\mathcal{E}_{\mathcal{I}}(G^*)$'s.

# 5 Experiments and implementation

We implement our verification algorithm and test its correctness on some well-known graphs such as cliques and trees, for which we know the exact verification number. In addition, we implement Algorithm 1 and compare its performance with other known atomic search algorithms [HG08, HB14, SKDV15, SMG+20] via the experimental setup of [SMG+20]: on synthetic graphs of varying sizes, we compare the runtime and total number of interventions performed compared to the verification number of the underlying DAG. In Appendix H, we provide the full experimental details and results of running various search algorithms (including ours) on different graphs. Qualitatively, our algorithm is competitive with the state-of-the-art search algorithms while being ∼10x faster in some experiments. Fig. 2 shows a subset of these results. We also investigated the impact of different $k$ values on the performance of our search algorithm in Appendix H. The implementations, along with entire experimental setup, are available at https://github.com/cxjdavin/verification-and-search-algorithms-for-causal-DAGs.

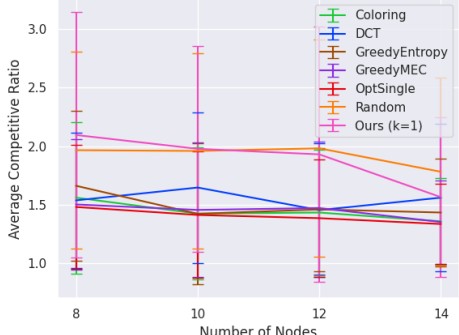 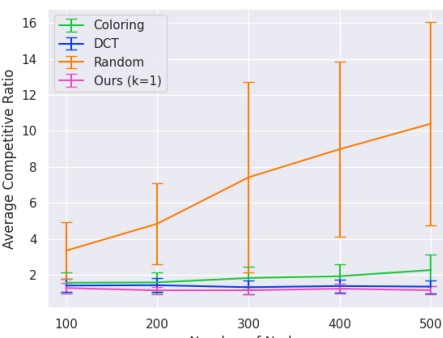

Figure 2: The figures show the average competitive ratios with respect to synthetic graphs of different node sizes. See Appendix H for details about how the synthetic graphs are generated and for details on the algorithms benchmarked. We also show the *maximum* competitive ratios in Appendix H.

**Conclusion, limitations, and societal impact**  Learning causal relationships is of fundamental importance to science and society in general. This is especially important when one wishes to correctly predict effects of making changes to a system for downstream tasks such as designing fair algorithms. In this work, we gave a complete understanding of the verification problem and an improved search algorithm under the standard causal inference assumptions (see Section 1). However, if our assumptions are violated by the data, then wrong causal conclusions may be drawn and possibly lead to unintended downstream consequences. Hence, it is of great interest to remove/weaken these assumptions while maintaining strong theoretical guarantees. A crucial limitation of this work is that we study an idealized setting with hard interventions and infinite samples while soft interventions may be more realistic in certain real-life scenarios (e.g. effects from parental vertices are not completely removed but only altered) and sample complexities play a crucial role when one has limited experimental budget (e.g. see [KJSB19] and [ABDK18] respectively).

## Acknowledgments and Disclosure of Funding

This research/project is supported by the National Research Foundation, Singapore under its AI Singapore Programme (AISG Award No: AISG-PhD/2021-08-013). KS was supported by a Stanford Data Science Scholarship, a Dantzig-Lieberman Research Fellowship and a Simons-Berkeley Research Fellowship. AB is partially supported by an NUS Startup Grant (R-252-000-A33-133) and an NRF Fellowship for AI (NRFFAI1-2019-0002). We would like to thank Themis Gouleakis, Dimitrios Myrisiotis, and Chandler Squires for valuable feedback and discussions. Part of this work was done while the authors were visiting the Simons Institute for the Theory of Computing.

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
