# A    Meek rules

Meek rules are a set of 4 edge orientation rules that are sound and complete with respect to any given set of arcs that has a consistent DAG extension [Mee95]. Given any edge orientation information, one can always repeatedly apply Meek rules till a fixed point to maximize the number of oriented arcs.

**Definition 23** (Consistent extension). A set of arcs is said to have a *consistent DAG extension* $\pi$ for a graph $G$ if there exists a permutation on the vertices such that (i) every edge $\{u, v\}$ in $G$ is oriented $u \to v$ whenever $\pi(u) < \pi(v)$, (ii) there is no directed cycle, (iii) all the given arcs are present.

**Definition 24** (The four Meek rules [Mee95], see Fig. 3 for an illustration).

**R1** Edge $\{a, b\} \in E$ is oriented as $a \to b$ if $\exists\, c \in V$ such that $c \to a$ and $c \not\sim b$.

**R2** Edge $\{a, b\} \in E$ is oriented as $a \to b$ if $\exists\, c \in V$ such that $a \to c \to b$.

**R3** Edge $\{a, b\} \in E$ is oriented as $a \to b$ if $\exists\, c, d \in V$ such that $d \sim a \sim c$, $d \to b \leftarrow c$, and $c \not\sim d$.

**R4** Edge $\{a, b\} \in E$ is oriented as $a \to b$ if $\exists\, c, d \in V$ such that $d \sim a \sim c$, $d \to c \to b$, and $b \not\sim d$.

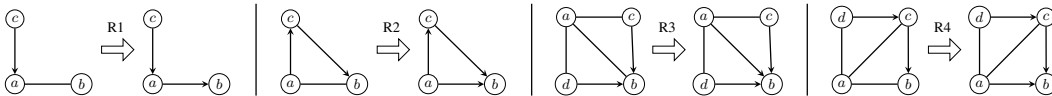

Figure 3: An illustration of the four Meek rules

There exists an algorithm [WBL21, Algorithm 2] that runs in $\mathcal{O}(d \cdot |E \cup A|)$ time and computes the closure under Meek rules, where $d$ is the degeneracy of the graph skeleton[13].

# B    Proof of Lemma 3

Lemma 1 of [HB14] actually considers a *single* additional intervention, but a closer look at their proof shows that the statement can be strengthened to allow for *multiple* additional interventions. In fact, the proof below will almost mimic the proof of [HB14, Lemma 1] except for some minor changes[14]. Note that we can drop the $\emptyset$ intervention in the statement since essential graphs are defined with the observational data provided. The proof relies on the definition of *strongly protected edges* and a characterization of $\mathcal{I}$-essential graphs from [HB12].

**Definition 25** (Strong protection; Definition 14 of [HB12]). Let $G = (V, E, A)$ be a (partially oriented) DAG and $\mathcal{I} \subseteq 2^V$ be an intervention set. An arc $a \to b$ is *strongly $\mathcal{I}$-protected* in $G$ if there is some intervention $S \in \mathcal{I}$ such that $|S \cap \{a, b\}| = 1$, or the arc $a \to b$ occurs in at least one of the following four configurations as an induced subgraph of $G$ (see Fig. 4):

1. There exists $c \in V$ such that $c \to a \to b$ and $c \not\sim b$.

2. There exists $c \in V$ such that $a \to b \leftarrow c$ and $c \not\sim a$.

3. There exists $c \in V$ such that $a \to c \to b$ and $a \to b$.

4. There exists $c, d \in V$ such that $a \sim c \to b$, $a \sim d \to b$, and $a \to b$.

**Theorem 26** (Characterization of $\mathcal{I}$-essential graphs; Theorem 18 of [HB12]). *A (partially oriented) DAG $\mathcal{E}(G)$ is an $\mathcal{I}$-essential graph of $G$ if and only if*

1. *$G$ is a chain graph.*

---

[13]A $d$-degenerate graph is an undirected graph in which every subgraph has a vertex of degree at most $d$. Note that the degeneracy of a graph is typically smaller than the maximum degree of the graph.

[14][HB14] considered whether the additional intervention $S \subseteq V$ separates a particular edge. In our proof, we change that argument to whether *some* intervention $S \in \mathcal{I}'$ separates that same edge. To argue that two graphs $G$ and $H$ are the same, one can show that $G \subseteq H$ and $H \subseteq G$. They only proved "one direction" and claim that the other holds by similar arguments. For completeness, we state *exactly* what are changes needed.

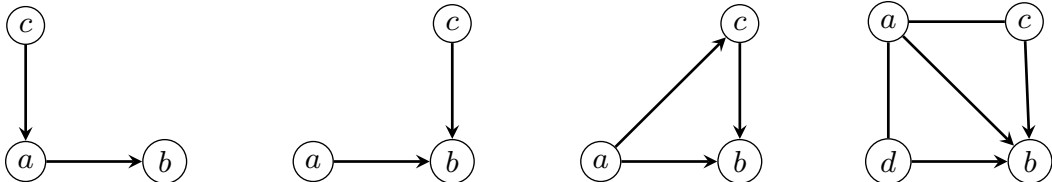

Figure 4: An illustration of the four configurations of strongly protected arc $a \to b$

    2. *For each chain component $H \in CC(G)$, $G[V(H)]$ is chordal.*

    3. *$G$ has no induced subgraph of the form $a \to b \sim c$.*

    4. *$G$ has no undirected edge $a \sim b$ whenever $\exists S \in \mathcal{I}$ such that $|S \cap \{a, b\}| = 1$.*

    5. *Every arc $a \to b$ in $G$ is strongly $\mathcal{I}$-protected.*

For simplicity, we say that an intervention $S \subseteq V$ *separates* an edge $a \sim b$ if $|S \cap \{a, b\}| = 1$ and that an intervention set $\mathcal{I} \subseteq 2^V$ *separates* an edge $a \sim b$ if it has an intervention that separates it.

**Lemma 3** (Modified lemma 1 of [HB14]). *Let $\mathcal{I} \subseteq 2^V$ be an intervention set. Consider the $\mathcal{I}$-essential graph $\mathcal{E}_\mathcal{I}(G^*)$ of some DAG $G^*$ and let $H \in CC(\mathcal{E}_\mathcal{I}(G^*))$ be one of its chain components. Then, for any additional interventional set $\mathcal{I}' \subseteq 2^V$ such that $\mathcal{I} \cap \mathcal{I}' = \emptyset$, we have*

$$\mathcal{E}_{\mathcal{I} \cup \mathcal{I}'}(G^*)[V(H)] = \mathcal{E}_{\{S \cap V(H) \,:\, S \in \mathcal{I}'\}}(G^*[V(H)]).$$

*Proof.* To shorten notation, define $G = \mathcal{E}_{\mathcal{I} \cup \mathcal{I}'}(G^*)[V(H)]$ and $G' = \mathcal{E}_{\{S \cap V(H) \,:\, S \in \mathcal{I}'\}}(G^*[V(H)])$. Since $G = (V, E, A)$ and $G' = (V', E', A')$ share the same skeleton (i.e. $V = V'$ and $E \cup A = E' \cup A'$) and must respect the same underlying DAG directions of $G^*$, it suffices to argue that $A \subseteq A'$ and $A' \subseteq A$ (i.e. they share the same set of directed arcs). Let $\pi$ be the topological ordering of the ground truth DAG $G^*$.

**Direction 1** ($A \subseteq A'$): Suppose there are directed arcs in $G$ that are undirected in $G'$. Let $a \to b$ be one such arc where $\pi(b)$ is *minimized*.

By property 5 of Theorem 26, $a \to b$ is strongly $(\mathcal{I} \cup \mathcal{I}')$-protected in $G$. If $\{S \cap V(H) \,:\, S \in \mathcal{I}'\}$ separates $a \sim b$, then $a \to b$ must be oriented in $G'$. Otherwise, let us consider the 4 configurations given by Definition 25:

    1. In $G$, there exists $c \in V$ such that $c \to a \to b$ and $c \not\sim b$. By minimality of $\pi(b)$, $c \to a$ must be oriented in $G'$. Thus, Meek rule R1 will orient $a \to c$.

    2. In $G$, there exists $c \in V$ such that $a \to b \leftarrow c$ and $c \not\sim a$. This is a v-structure in $G^*$ and so $a \to b$ would also be oriented in $G'$.

    3. In $G$, there exists $c \in V$ such that $a \to c \to b$ and $a \to b$. By minimality of $\pi(b)$, $a \to c$ must be oriented in $G'$. Then, if $a \to b$ is *not* directed in $G$', we will have a directed cycle of the form $a \to c \sim b \sim a$ (regardless of whether the edge $b \sim c$ is directed). By property 1 of Theorem 26, $G'$ is a chain graph and *cannot* have such a directed cycle. Therefore, $a \to b$ must be oriented in $G'$.

    4. In $G$, there exists $c, d \in V$ such that $a \sim c \to b$, $a \sim d \to b$, and $a \to b$. We cannot have $c \to a \leftarrow d$ otherwise such a v-struct will prevent this configuration from occurring. Without loss of generality, $a \to c$. Then, we can apply the argument of the third configuration on the subgraph induced by $\{a, b, c\}$ to conclude that $a \to b$ is also oriented in $G'$.

**Direction 2** ($A' \subseteq A$): Repeat the *exact* same argument but perform the following 2 swaps:

    1. Swap the roles of $G$ and $G'$

    2. Swap the roles of $(\mathcal{I} \cup \mathcal{I}')$ and $\{S \cap V(H) \,:\, S \in \mathcal{I}'\}$       $\square$

# C   Further analysis of the standing windmill essential graph

In this section, we show that *all* DAGs in the standing windmill essential graph requires at least 3 and at most 4 atomic interventions.

By Theorem 11, we know that the optimal number of atomic interventions needed to verify any graph is the size of the minimum vertex cover of its oriented edges. To explore the space of DAGs in the essential graph, we will perform covered edge reversals (as justified by Lemma 7).

Consider the DAG $G^*$ with MEC $[G^*]$ and the standing windmill essential graph $\mathcal{E}(G^*)$ in Fig. 5. Starting from $G^*$, if we fix the arc direction $h \to a$, then reversing any arc (possibly multiple times) from the set $\{b \sim c, d \sim e, f \sim g\}$ does *not* change the covered edge status of any edge (i.e. the covered edges remain exactly the same 4 edges) and thus the size of the minimum vertex cover remains unchanged. Meanwhile, reversing $a \sim h$ in $G^*$ yields the graph $G_1$. Fixing the arc direction $a \to h$, we observe that the three sets of edges $\{a \sim b, a \sim c, b \sim c\}$, $\{a \sim d, a \sim e, d \sim e\}$, and $\{a \sim f, a \sim g, f \sim g\}$ are symmetric. Furthermore, if we flip one of the edges from $\{a \sim b, a \sim d, a \sim f\}$ from $G_1$ (or $\{a \sim c, a \sim d, a \sim f\}$ from $G_4$), then all other two $a \to \cdot$ arcs are no longer covered edges. So, it suffices to study what happens when we only reverse arc directions in one of these sets: $\{a \sim b, a \sim c, b \sim c\}$, $\{a \sim d, a \sim e, d \sim e\}$, and $\{a \sim f, a \sim g, f \sim g\}$. The graphs $G_1$ to $G_6$ illustrate all possible cases when we fix $a \to h$ and only reverse edges in the set $\{a \sim b, a \sim c, b \sim c\}$. We see that $\nu_1(G^*) = \nu_1(G_1) = \nu_1(G_4) = 4$ and $\nu_1(G_2) = \nu_1(G_3) = \nu_1(G_5) = \nu_1(G_6) = 3$. Thus, we can conclude that $\min_{G \in [G^*]} \nu_1(G) = 3$ and $\max_{G \in [G^*]} \nu_1(G) = 4$.

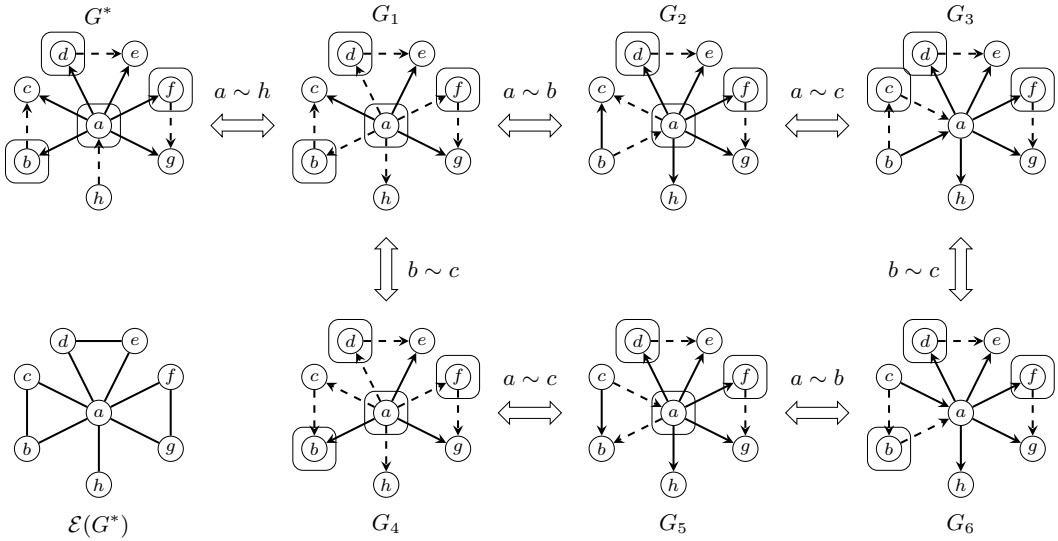

Figure 5: A DAG $G^*$ with its essential graph $\mathcal{E}(G^*)$ and some of the graphs $G \in [G^*]$. In each DAG, dashed arcs are covered edges and the boxed vertices represent a minimum vertex cover.

# D   Upper and (worst case) lower bounds for fully orienting a DAG from its essential graph using ideal interventions

Let us briefly distinguish the various problem settings before summarizing the state of the art results.

**Intervention size**   Since interventions are expensive, natural restrictions on the size of any intervention $S \in \mathcal{I}$ has been studied. Bounded size interventions enforce that an upper bound of $|S| \le k$ always while unbounded size interventions allow $k$ to be as large as $n/2$. Note that it does not make sense to intervene on a set $S$ with $|S| > n/2$ since intervening on $\bar{S}$ yields the same information while being a strictly smaller interventional set. Atomic interventions are a special case where $k = 1$.

**Adaptivity**  A passive/non-adaptive/simultaneous algorithm is one which, given an essential graph $\mathcal{E}(G^*)$, decides a *set* of interventions without looking at the outcomes of the interventions. Meanwhile, active/adaptive algorithms can provide a *sequence* of interventions one-at-a-time, possibly using any information gained from the outcomes of earlier chosen interventions.

**Determinism**  An algorithm is deterministic if it always produces the same output given the same input. Meanwhile, randomized algorithms produces an output from a distribution. Analyses of randomized algorithms typically involve probabilistic arguments and their performance is measured in expectation with probabilistic success[15]. The ability to use random bits (e.g. outcome of coin flips) is very powerful and may allow one to circumvent known deterministic lower bounds.

**Special graph classes**  Two graph classes of particular interest are cliques and trees. If $CC(\mathcal{E}(G^*))$ is a clique, then all $\binom{n}{2}$ edges are present and fully orienting the clique is equivalent to finding the unique valid permutation on the vertices. As such, cliques are often used to prove worst case lower bounds. Meanwhile, if $CC(\mathcal{E}(G^*))$ is a tree, then there must be a unique root (else there will be v-structures) and it suffices[16] to intervene on the root node to fully orient the tree.

Table 1 and Table 2 summarize some existing upper (sufficient) and lower (worst case necessary) bounds on the size ($|\mathcal{I}|$, or $\mathbb{E}[|\mathcal{I}|]$ for randomized algorithms) of intervention sets that fully orient a given essential graph. These lower bounds are "worst case" in the sense that there exists a graph, typically a clique, which requires the stated number of interventions. Observe that there are settings where adaptivity and randomization strictly improves the number of required interventions.

| Size | Adaptive | Randomized | Graph | Upper bound | Reference |
|------|----------|------------|-------|-------------|-----------|
| 1 | ✗ | ✗ | General | $n - 1$ | [EGS06] |
| 1 | ✗ | ✓ | General | $\frac{2}{3}n - \frac{1}{3}$ for $n > 3$ | [Ebe10] |
| 1 | ✓ | ✗ | Tree | $\mathcal{O}(\log n)$ | [SKDV15] |
| 1 | ✓ | ✗ | Tree | $\lceil \log n \rceil$ | [GKS$^+$19] |
| $\leq k$ | ✗ | ✗ | General | $(\frac{n}{k} - 1) + \frac{n}{2k} \log_2 k$ | [EGS12] |
| $\leq k$ | ✓ | ✗ | Tree | $\lceil \log_{k+1} n \rceil$ | [GKS$^+$19] |
| $\leq k$ | ✓ | ✓ | Clique | $\mathcal{O}(\frac{n}{k} \log \log k)$ | [SKDV15] |
| $\infty$ | ✗ | ✗ | General | $\log_2 n$ | [EGS12] |
| $\infty$ | ✗ | ✗ | General | $\lceil \log_2(\omega(\mathcal{E}(G))) \rceil$ | [HB14] |
| $\infty$ | ✗ | ✓ | General | $\mathcal{O}(\log \log n)$ | [HLV14] |

Table 1: Upper bounds on the size ($|\mathcal{I}|$, or $\mathbb{E}[|\mathcal{I}|]$ for randomized algorithms) of the intervention set sufficient to fully orient a given essential graph $\mathcal{E}(G)$. The first three columns indicate the setting which the algorithm operates in terms of intervention size, adaptivity, and randomness. The fourth column indicate whether the algorithm is for special graph classes. Roughly speaking, the algorithm has *more power as we move down the rows* since it can use larger intervention sets, be adaptive, utilize randomization, and possibly only work on special graph classes.

# E  Verification

## E.1  Properties of covered edges

**Lemma 17** (Properties of covered edges)**.**

1. *Let $H$ be the edge-subgraph induced by covered edges of a DAG $G$. Then, every vertex in $H$ has at most one incoming edge and thus $H$ is a forest of directed trees.*

2. *If a DAG $G$ is a clique on $n \geq 3$ vertices $v_1, v_2, \ldots, v_n$ with $\pi(v_1) < \pi(v_2) < \ldots < \pi(v_n)$, then $v_1 \to v_2, \ldots, v_{n-1} \to v_n$ are the covered edges of $G$.*

---

[15]Typically, they will be shown to succeed with high probability in $n$: as the size of the graph $n$ increases, the failure probability decays quickly in the form of $n^{-c}$ for some constant $c > 1$.

[16]This will later be obvious through the lenses of covered edges: all covered edges are incident to the root.

| Size | Adaptive | Randomized | Lower bound | Reference |
|:---:|:---:|:---:|:---:|:---:|
| 1 | ✗ | ✓ | $\frac{2}{3}n - \frac{1}{3}$ for $n > 3$ | [Ebe10] |
| 1 | ✓ | ✗ | $n - 1$ | [EGS06] |
| $\leq k$ | ✗ | ✗ | $(\frac{n}{k} - 1) + \frac{n}{2k} \log_2 k$ | [EGS12] |
| $\leq k$ | ✓ | ✓ | $\frac{n}{2k}$ | [SKDV15] |
| $\infty$ | ✗ | ✗ | $\log_2 n$ | [EGS12] |
| $\infty$ | ✗ | ✓ | $\Omega(\log \log n)$ | [HLV14] |
| $\infty$ | ✓ | ✓ | $\lceil \log_2(\omega(\mathcal{E}(G)) \rceil$ | [HB14] |

Table 2: Lower bounds on the size ($|\mathcal{I}|$, or $\mathbb{E}[|\mathcal{I}|]$ for randomized algorithms) of the intervention set necessary to fully orient a given essential graph $\mathcal{E}(G)$. The first three columns indicate the setting which the algorithm operates in terms of intervention size, adaptivity, and randomness. Roughly speaking, the setting becomes *easier as we move down the rows* so the lower bounds are *stronger as we move down the rows*. On cliques, [SKDV15] also showed that $\geq n/2$ vertices must be intervened.

3. *If $u \to v$ is a covered edge in a DAG $G$, then $u$* cannot *be a sink of any maximal clique of $G$.*

*Proof.*

1. Suppose, for a contradiction, that there exists some vertex $w$ with two incoming covered edges $u \to w \leftarrow v$. For $u \to w$ to be covered, we must have $v \to u$. Similarly, for $v \to w$ to be covered, we must have $u \to v$. However, we cannot simultaneously have both $u \to v$ and $v \to u$, as it would lead to a contradiction as $G$ is a DAG. Furthermore, since $G$ is acyclic, it implies that $H$ must also be acyclic. Therefore $H$ is a forest of directed trees.

2. Let $A = \{v_1 \to v_2, v_2 \to v_3, \ldots, v_{n-1} \to v_n\}$ be the set of arcs of interest. For any arc $v_i \to v_{i+1} \in A$, one can check that they share the same parents by the topological ordering $\pi$. Consider an arbitrary arc $v_i \to v_j \notin A$. Since $v_i \to v_j \notin A$, there exists $v_k \in V$ such that $\pi(v_i) < \pi(v_k) < \pi(v_j)$. Then, since $G$ is a clique, we must have $v_i \to v_k \to v_j$ and so $v_i \to v_j$ *cannot* be covered since $v_k \in Pa(v_j) \setminus \{v_i\}$ but $v_k \notin Pa(v_i) \setminus \{v_j\}$.

3. Suppose, for a contradiction, that $u$ is a sink of some maximal clique $K_h$ of size $h$ and $u \to v$ is a covered edge. Then, we must have $Pa(v) \setminus \{u\} = Pa(u) \setminus v$. However, that means that $V(K_h) \cup \{v\}$ is a clique of size $h + 1$. Thus, $K_h$ was not a maximal clique. Contradiction.

$\square$

### E.2 Characterization via separation of covered edges

**Lemma 27** (Necessary). *Fix an essential graph $\mathcal{E}(G^*)$ and $G \in [G^*]$. If $\mathcal{I} \subseteq 2^V$ is a verifying set, then $\mathcal{I}$ separates all unoriented covered edge $u \sim v$ of $G$.*

*Proof.* Let $u \to v$ be an arbitrary unoriented covered edge in $\mathcal{E}(G^*)$ and $\mathcal{I}$ be an intervention set where $u$ and $v$ are *never* separated by any $S \in \mathcal{I}$. Then, interventions will not orient $u \to v$ and we can only possibly orient it via Meek rules. We check that all four Meek rules will *not* orient $u \to v$:

**(R1)** For R1 to trigger, we need to have $w \to u \to v$ and $w \not\sim v$ for some vertex $w \in V \setminus \{u, v\}$. However, such a vertex $w$ will imply that $u \to v$ is *not* a covered edge.

**(R2)** For R2 to trigger, we need to have $u \to w \to v$ for some $w \in V \setminus \{u, v\}$. However, such a vertex $w$ will imply that $u \to v$ is *not* a covered edge.

**(R3)** For R3 to trigger, we must have $w \sim u \sim x$, $w \to v \leftarrow x$, and $w \not\sim x$ for some $w, x \in V \setminus \{u, v\}$. Since $u \to v$ is a covered edge, we must have $w \to u \leftarrow x$. This implies that both $w \to u \leftarrow x$ appear as v-structures in $\mathcal{E}(G^*)$ and thus R3 will not trigger.

**(R4)** For R4 to trigger, we must have $w \sim u \sim x$, $w \to x \to v$, and $w \not\sim v$ for some $w, x \in V \setminus \{u, v\}$. Since $u \to v$ is covered, we must have $x \to u$. To avoid directed cycles, it must be the case that $w \to u$. However, this implies that $u \to v$ is *not* covered since $w \to u$ while $w \not\sim v$.

Therefore, $\mathcal{I}$ *cannot* be a verifying set if $u$ and $v$ are *never* separated by any $S \in \mathcal{I}$. $\qquad\square$

**Lemma 28** (Sufficient). *Fix an essential graph $\mathcal{E}(G^*)$ and $G \in [G^*]$. If $\mathcal{I} \subseteq 2^V$ is an intervention set that separates every unoriented covered edge $u \sim v$ of $G$, then $\mathcal{I}$ is a verifying set.*

*Proof.* Let $\mathcal{I}$ be an arbitrary intervention set such that every unoriented covered edge $u \sim v$ of $G$ has an set $S \in \mathcal{I}$ that separates $u$ and $v$. Fix an arbitrary valid vertex permutation $\pi : V \to [n]$ of $G$. For any $i \in [n]$, define $V_i = \{\pi^{-1}(1), \ldots, \pi^{-1}(i)\} \subseteq V$ as the $i$ smallest vertices according to $\pi$'s ordering. We argue that any unoriented edges in $\mathcal{E}(G^*)[V_i]$ will be oriented by $\mathcal{I}$ by performing induction on $i$.

**Base case** ($i = 1$): There are no edges in $G[V_1]$ so $\mathcal{E}(G^*)[V_1]$ is trivially fully oriented.

**Inductive case** ($i > 1$): Suppose $v = \pi^{-1}(i)$. By induction hypothesis, $\mathcal{E}(G^*)[V_{i-1}]$ is fully oriented so any unoriented edge in $\mathcal{E}(G^*)[V_i]$ must have the form $u \to v$, where $\pi(u) < \pi(v)$. For any $u \to v$ is an unoriented covered edge in $\mathcal{E}(G^*)[V_i]$, there will be an intervention $S \in \mathcal{I}$ that separates $u$ and $v$ (or both), and hence orient $u \to v$.

Suppose, for a contradiction, that there exists unoriented edges in $\mathcal{E}(G^*)[V_i]$ that are *not* covered edges. Let $u \to v$ be the unoriented edge where $\pi(u)$ is *maximized*. Then, one of the two cases must occur:

**Case 1** ($u \to v$ and $\exists w \in V_i$ such that $w \to u$ and $w \not\to v$) Since $\pi(v) > \pi(w)$, we must have $w \not\sim v$. By induction, $w \to u$ will be oriented. So, R1 orients $u \to v$.

**Case 2** ($u \to v$ and $\exists w \in V_i$ such that $w \to v$ and $w \not\to u$) If $w \not\sim u$, then $u \to v \leftarrow w$ is a v-structure and $u \to v$ would have been oriented. If $w \sim u$, then we must have $u \to w$ and $\pi(u) < \pi(w)$. By induction, $u \to w$ will be oriented. Since $\pi(u) < \pi(w)$ and $\pi(u)$ is maximized out of all possible unoriented edges in $\mathcal{E}(G^*)[V_i]$ involving $v$, $w \to v$ must be an oriented edge and will be oriented by $\mathcal{I}$. So, R2 orients $u \to v$.

In either case, $u \to v$ will be oriented. Contradiction. $\qquad\square$

Combining Lemma 27 and Lemma 28 gives the following characterization of verifying sets.

**Theorem 9.** *Fix an essential graph $\mathcal{E}(G^*)$ and $G \in [G^*]$. An intervention set $\mathcal{I}$ is a verifying set for $G$ if and only if $\mathcal{I}$ is a set that separates every covered edge of $G$ that is unoriented in $\mathcal{E}(G^*)$.*

### E.3 Solving the verification problem

**Corollary 10.** *Given an essential graph $\mathcal{E}(G^*)$ of an unknown ground truth DAG $G^*$ and a causal DAG $G \in [G^*]$, we can test if $G \overset{?}{=} G^*$ by intervening on any verifying set of $G$. Furthermore, in the worst case, any algorithm that correctly resolves $G \overset{?}{=} G^*$ needs at least $\nu(G)$ interventions.*

*Proof.* Using Theorem 9, we know that the minimal verifying set for $G$ is the smallest possible set of interventions $\mathcal{I}$ such that *all* covered edges of $G$ is separated by some intervention $S \in \mathcal{I}$. If the graph is fully oriented after intervening on all $S \in \mathcal{I}$, then it must be the case that $G = G^*$. Otherwise, we will either detect that some edge orientation disagrees with $G$ or there remains some unoriented edge at the end of all our interventions. In the first case, we trivially conclude that $G \neq G^*$. In the second case, Theorem 9 tells us that any such unoriented edge must be an unoriented covered edge of $G^*$ (which was not an unoriented cover edge of $G^A$) and so we can also conclude that $G \neq G^*$.

Suppose, for a contradiction, that some algorithm $\mathcal{A}$ uses strictly less than $\nu(G)$ interventions to verify a graph $G$. Then, there exists at least one covered edge $u \to v$ in $G$ is not separated by the interventions used. Define $G'_1 = G$ and $G'_2$ as $G$ with this covered edge reversed (i.e. $v \to u$ instead). Note that $G'_2$ is also a DAG in the same MEC due to Lemma 7. We see that $\mathcal{A}$ *cannot* distinguish

between $G_1'$ and $G_2'$ and thus cannot correctly output $G = G'$ or $G \neq G'$ respectively. This is a contradiction, i.e. at least $\nu(G)$ interventions are needed in the worst case. $\square$

### E.4 Efficient optimal verification via atomic interventions

**Theorem 11.** *Fix an essential graph $\mathcal{E}(G^*)$ and $G \in [G^*]$. An atomic intervention set $\mathcal{I}$ is a minimal sized verifying set for $G$ if and only if $\mathcal{I}$ is a minimum vertex cover of unoriented covered edges of $G$. A minimal sized atomic verifying set can be computed in polynomial time.*

*Proof.* For $|S| = 1$, we see that $\mathcal{I}$ separates every unoriented covered edge in $\mathcal{E}(G)$ if and only if the set $\cup_{S \in \mathcal{I}} S$ is a vertex cover of the unoriented covered edges in $\mathcal{E}(G)$. Lemma 17 tells us that the edge-induced subgraph on covered edges of $G$ is a forest. Thus, one can perform the standard dynamic programming algorithm to compute the minimum vertex cover on each tree. $\square$

### E.5 Efficient near-optimal verification via bounded size interventions

We first prove a simple lower bound on the minimum number of non-atomic bounded size interventions (i.e. $|S| \leq k$) needed for verification and then show how to adapt a minimal atomic verifying set to obtain a near-optimal bounded size verifying set.

**Lemma 29.** *Fix an essential graph $\mathcal{E}(G^*)$ and $G \in [G^*]$. Suppose $\mathcal{I}$ is an arbitrary bounded size intervention set. Intervening on vertices in $\cup_{S \in \mathcal{I}} S$ one at a time, in an atomic fashion, can only increase the number of separated covered edges of $G$.*

*Proof.* Consider an arbitrary covered edge $u \sim v$ that was seprated by some intervention $S \in \mathcal{I}$. This means that $|\{u, v\} \cap S| = 1$. Without loss of generality, suppose $u \in S$. Then, when we intervene on $u$ in an atomic fashion, we would also separate the edge $u \sim v$. $\square$

**Lemma 30.** *Fix an essential graph $\mathcal{E}(G^*)$ and $G \in [G^*]$. If $\nu_1(G) = \ell$, then $\nu_k(G) \geq \lceil \frac{\ell}{k} \rceil$.*

*Proof.* A bounded size intervention set of size strictly less than $\lceil \frac{\ell}{k} \rceil$ involves strictly less than $\ell$ vertices. By Lemma 29, such an intervention set cannot be a verifying set. $\square$

**Lemma 31.** *Fix an essential graph $\mathcal{E}(G^*)$ and $G \in [G^*]$. If $\nu_1(G) = \ell$, then there exists a polynomial time algorithm that computes a bounded size verifying set $\mathcal{I}$ of size $|\mathcal{I}| \leq \lceil \frac{\ell}{k} \rceil + 1$.*

*Proof.* Consider any atomic verifying set $\mathcal{I}$ of $G$ of size $\ell$. By Lemma 17, the edge-induced subgraph on covered edges of $G$ is a forest and is thus 2-colorable.

Split the vertices in $\mathcal{I}$ into partitions according to the 2-coloring. By construction, vertices belonging in the same partite will *not* be adjacent and thus choosing them together to be in an intervention $S$ will *not* reduce the number of separated covered edges. Now, form interventions of size $k$ by greedily picking vertices in $\mathcal{I}$ within the same partite. For the remaining unpicked vertices (strictly less than $k$ of them), we form a new intervention with them. Repeat the same process for the other partite.

This greedy process forms groups of size $k$ and at most 2 groups of sizes, one from each partite. Suppose that we formed $z$ groups of size $k$ in total and two "leftover groups" of sizes $x$ and $y$, where $0 \leq x, y < k$. Then, $\ell = z \cdot k + x + y$, $\frac{\ell}{k} = z + \frac{x+y}{k}$, and we formed at most $z + 2$ groups. If $0 \leq x + y < k$, then $\lceil \frac{\ell}{k} \rceil = z + 1$. Otherwise, if $k \leq x + y < 2k$, then $\lceil \frac{\ell}{k} \rceil = z + 2$. In either case, we use at most $\lceil \frac{\ell}{k} \rceil + 1$ interventions, each of size $\leq k$.

One can compute a bounded size intervention set efficiently because the following procedures can all be run in polynomial time: (i) checking if each edge is a covered edge; (ii) computing a minimum vertex cover on a tree; (iii) 2-coloring a tree; (iv) greedily grouping vertices into sizes $\leq k$. $\square$

Theorem 12 follows by combining Lemma 30 and Lemma 31.

**Theorem 12.** *Fix an essential graph $\mathcal{E}(G^*)$ and $G \in [G^*]$. If $\nu_1(G) = \ell$, then $\nu_k(G) \geq \lceil \frac{\ell}{k} \rceil$ and there exists a polynomial time algo. to compute a bounded size intervention set $\mathcal{I}$ of size $|\mathcal{I}| \leq \lceil \frac{\ell}{k} \rceil + 1$.*

Observe that there exists graphs and values $k$ such that the optimal bounded size verifying set requires at least $\lceil \frac{\ell}{k} \rceil + 1$, and thus our upper bound is tight in the worst case: Fig. 6 shows there exists a family of graphs (and values $k$) such that the optimal bounded size verifying set requires $\lceil \frac{\ell}{k} \rceil + 1$. However, we do not have a proof that Theorem 12 is optimal (or counter example that it is not).

*Conjecture* 32. The construction of bounded size verifying set given in Theorem 12 is optimal.

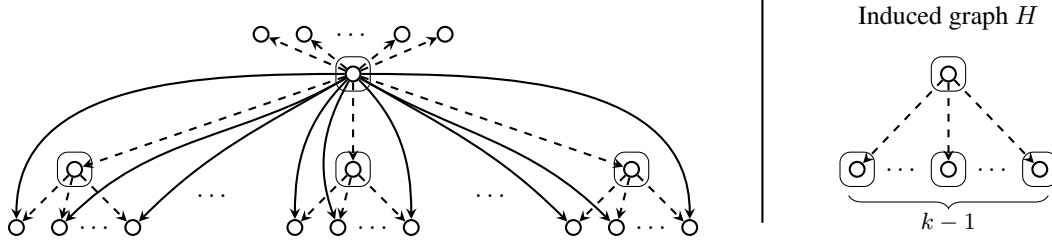

Figure 6: A DAG with its covered edges given in dashed arcs. The edge-induced subgraph of the covered edges is a tree and the minimum vertex cover is all the non-leaf vertices (the boxed vertices) of size $\ell$. Denote the graph induced by the boxed vertices by $H$. Now consider the star graph $H$ on $\ell = k$ nodes with $k - 1$ leaves. All the leaf nodes can be put in the same intervention without affecting the separation of any covered edges. However, including the root with any of the leaf nodes in a same intervention will cause covered edges to be unseparated. Thus, using bounded size interventions of size at most $k$, verifying such a DAG requires at least $\lceil \frac{\ell}{k} \rceil + 1 = 2$ interventions.

### E.6 Generalization to minimum cost verifying sets with additive cost structures

Consider an essential graph which is a star graph on $n$ nodes where the leaves have cost 1 and the root has cost significantly larger than $n$. For atomic verifying sets, we see that the *minimum cost* verifying set to intervene on the leaves one at a time while the *minimum size* verifying set is to simply intervene on the root. Since one may be more preferred over the other, depending on the actual real-life situation, we propose to find a verifying set $\mathcal{I}$ which minimizes

$$\alpha \cdot w(\mathcal{I}) + \beta \cdot |\mathcal{I}| \qquad \text{where } \alpha, \beta \geq 0 \tag{1}$$

so as to explicitly trade-off between the cost and size of the intervention set. This objective also naturally allows the constraint of bounded size interventions by restricting $|S| \leq k$ for all $S \in \mathcal{I}$.

**Theorem 13.** *Fix an essential graph $\mathcal{E}(G^*)$ and $G \in [G^*]$. An atomic verifying set $\mathcal{I}$ for $G$ that minimizes Eq. (1) can be computed in polynomial time.*

*Proof.* By Theorem 9 and accounting for Eq. (1), we need to compute a *weighted* minimum vertex cover in the edge-induced subgraph on covered edges of $G$. Efficiency is implied by Lemma 17. □

For bounded size interventions, we show that the ideas in Appendix E.5 translate naturally to give a near-optimal minimal generalized cost verifying set. To prove our lower bound, we first consider an optimal atomic verifying set for a slightly different objective from Eq. (1).

**Lemma 33.** *Fix an essential graph $\mathcal{E}(G^*)$ and $G \in [G^*]$. Let $\mathcal{I}_A$ be an atomic verifying set for $G$ that minimizes $\alpha \cdot w(\mathcal{I}_A) + \frac{\beta}{k} \cdot |\mathcal{I}_A|$ and $\mathcal{I}_B$ be a bounded size verifying set for $G$ that minimizes Eq. (1). Then, $\alpha \cdot w(\mathcal{I}_A) + \frac{\beta}{k} \cdot |\mathcal{I}_A| \leq \alpha \cdot w(\mathcal{I}_B) + \beta \cdot |\mathcal{I}_B|$.*

*Proof.* Let $\mathcal{I} = \sum_{S \in \mathcal{I}_B} S$ be the atomic verifying set derived from $\mathcal{I}_B$ by treating each vertex as an atomic intervention. Clearly, $w(\mathcal{I}_B) \geq w(\mathcal{I})$ and $k \cdot |\mathcal{I}_B| \geq |\mathcal{I}|$. So,

$$\alpha \cdot w(\mathcal{I}_B) + \beta \cdot |\mathcal{I}_B| \geq \alpha \cdot w(\mathcal{I}) + \frac{\beta}{k} \cdot |\mathcal{I}| \geq \alpha \cdot w(\mathcal{I}_A) + \frac{\beta}{k} \cdot |\mathcal{I}_A|$$

since $\mathcal{I}_A = \text{argmin}_{\text{atomic verifying set } \mathcal{I}'} \left\{ \alpha \cdot w(\mathcal{I}') + \frac{\beta}{k} \cdot |\mathcal{I}'| \right\}$. □

**Theorem 14.** *Fix an essential graph $\mathcal{E}(G^*)$ and $G \in [G^*]$. Suppose the optimal bounded size intervention set that minimizes Eq. (1) costs $OPT$. Then, there exists a polynomial time algorithm that computes a bounded size intervention set with total cost $OPT + 2\beta$.*

*Proof.* Let $\mathcal{I}_A$ be an atomic verifying set for $G$ that minimizes $\alpha \cdot w(\mathcal{I}_A) + \frac{\beta}{k} \cdot |\mathcal{I}_A|$ and $\mathcal{I}_B$ be a bounded size verifying set for $G$ that minimizes Eq. (1). Using the polynomial time greedy algorithm in Lemma 31, we construct bounded size intervention set $\mathcal{I}$ by greedily grouping together atomic interventions from $\mathcal{I}_A$. Clearly, $w(\mathcal{I}) = w(\mathcal{I}_A)$ and $|\mathcal{I}| \leq \lceil \frac{|\mathcal{I}_A|}{k} \rceil + 1$. So,

$$\alpha \cdot w(\mathcal{I}) + \beta \cdot |\mathcal{I}| \leq \alpha \cdot w(\mathcal{I}_A) + \beta \cdot \left( \left\lceil \frac{|\mathcal{I}_A|}{k} \right\rceil + 1 \right) \leq \alpha \cdot w(\mathcal{I}_B) + \beta \cdot |\mathcal{I}_B| + 2\beta = OPT + 2\beta$$

where the second inequality is due to Lemma 33. □

## F Search

We begin by proving a strengthened version of [SMG+20]'s lower bound.

Note that we will be discussing only atomic interventions in Lemma 21, so notation such as $\mathcal{I} \cap V(H)$ makes sense for sets $\mathcal{I}, V(H) \subseteq V$.

**Definition 34** (Moral DAG, Definition 3 of [SMG+20]). *A graph $G$ is a moral DAG if its essential graph only has a single chain component. That is, after removing directed edges, there is only one single connected component remaining.*

**Lemma 35** (Lemma 6 of [SMG+20]). *Let $G$ be a moral DAG. Then, $\nu_1(G) \geq \lfloor \frac{\omega(skeleton(G))}{2} \rfloor$.*

**Lemma 21.** *Fix an essential graph $\mathcal{E}(G^*)$ and $G \in [G^*]$. Then,*

$$\nu_1(G) \geq \max_{\mathcal{I} \subseteq V} \sum_{H \in CC(\mathcal{E}_\mathcal{I}(G^*))} \left\lfloor \frac{\omega(H)}{2} \right\rfloor$$

*Proof.* Consider be an arbitrary set of atomic interventions $\mathcal{I} \subseteq V$ and the resulting $\mathcal{I}$-essential graph $\mathcal{E}_\mathcal{I}(G^*)$. Let $H \in CC(\mathcal{E}_\mathcal{I}(G^*))$ be an arbitrary chain component.

Let $\mathcal{I}' \subseteq V$ be an arbitrary atomic verifying set of $G$. Then, $\mathcal{E}_{\mathcal{I}'}(G^*) = G$ and thus $\mathcal{E}_{\mathcal{I}'}(G^*)[V(H)] = G[V(H)]$. Then,

$$\mathcal{E}_{(\mathcal{I}' \setminus \mathcal{I}) \cap V(H)}(G[V(H)]) = \mathcal{E}_{\mathcal{I} \cup (\mathcal{I}' \setminus \mathcal{I})}(G)[V(H)] = \mathcal{E}_{\mathcal{I}'}(G)[V(H)] = G[V(H)]$$

where the first equality is due to Lemma 3 and the last equality is because $\mathcal{I}'$ is a verifying set of $G$. So, $(\mathcal{I}' \setminus \mathcal{I}) \cap V(H)$ is a verifying set for $G[V(H)]$, and so is $\mathcal{I}' \cap V(H)$. Thus, by minimality of $\nu_1$, we have $\nu_1(G[V(H)]) \leq |\mathcal{I}' \cap V(H)|$ for *any* atomic verifying set $\mathcal{I}' \subseteq V$ of $G$.

Since $H \in CC(\mathcal{E}_\mathcal{I}(G^*))$, the graph $G[V(H)]$ is a moral DAG. Since $H$ is a subgraph of $G[V(H)]$, $\omega(H) \leq \omega(G[V(H)])$. Thus, by Lemma 35, we have $\nu_1(G[V(H)]) \geq \lfloor \frac{\omega(G[V(H)])}{2} \rfloor \geq \lfloor \frac{\omega(H)}{2} \rfloor$.

Now, suppose $\mathcal{I}^*$ is a minimal size verifying set of $G$. Then,

$$\nu_1(G) = |\mathcal{I}^*| \geq \sum_{H \in CC(\mathcal{E}_\mathcal{I}(G^*))} |\mathcal{I}^* \cap V(H)| \geq \sum_{H \in CC(\mathcal{E}_\mathcal{I}(G^*))} \nu_1(G[V(H)]) \geq \sum_{H \in CC(\mathcal{E}_\mathcal{I}(G^*))} \left\lfloor \frac{\omega(H)}{2} \right\rfloor$$

The claim follows by taking the maximum over all possible atomic interventions $\mathcal{I} \subseteq V$. □

For convenience, we reproduce Algorithm 1 below.

**Lemma 36.** *Algorithm 1 terminates after at most $\mathcal{O}(\log n)$ iterations.*

*Proof.* Consider an arbitrary iteration $i$ and chain component $H$ with 1/2-clique separator $K_H$.

Edges within $K_H$ will be fully oriented by Step 6. We now argue that any edge with exactly one endpoint in $K_H$ will be oriented by Algorithm 1. For $k'_H = 1$, the algorithm intervenes on all nodes in $K_H$ and thus such edges will be oriented trivially. For $k'_H > 1$, the additional $\leq |K_H|/k'_H$

---

**Algorithm 1** Search algorithm via graph separators.

---

1: **Input**: Essential graph $\mathcal{E}(G^*)$, intervention size $k$. **Output**: A fully oriented graph $G \in [G^*]$.
2: Initialize $i = 0$ and $\mathcal{I}_0 = \emptyset$.
3: **while** $\mathcal{E}_{\mathcal{I}_i}(G^*)$ still has undirected edges **do**
4:      For each $H \in CC(\mathcal{E}_{\mathcal{I}_i}(G^*))$ of size $|H| \geq 2$, find a 1/2-clique separator $K_H$ using
     Theorem 20. Define $Q = \{K_H\}_{H \in CC(\mathcal{E}_{\mathcal{I}_i}(G^*)), |H| \geq 2}$ as the union of clique separator nodes.
5:      **if** $k = 1$ or $|Q| = 1$ **then** Define $C_i = Q$ as an atomic intervention set.
6:      **else** Define $k' = \min\{k, |Q|/2\}$, $a = \lceil |Q|/k' \rceil \geq 2$, and $\ell = \lceil \log_a n \rceil$. Compute labelling
     scheme of [SKDV15, Lemma 1] on $Q$ with $(|Q|, k', a)$, and define $C_i = \{S_{x,y}\}_{x \in [\ell], y \in [a]}$,
     where $S_{x,y} \subseteq Q$ is the subset of vertices whose $x^{th}$ letter in the label is $y$.
7:      Update $i \leftarrow i + 1$, intervene on $C_i$ to obtain $\mathcal{E}_{\mathcal{I}_i}(G^*)$, and update $\mathcal{I}_i \leftarrow \mathcal{I}_{i-1} \cup C_i$.
8: **end while**

---

interventions *after* the bounded size intervention strategy of [SKDV15] ensures that *every* edge with exactly one endpoint in $K_H$ will be separated. Thus, after each iteration, the only remaining unoriented edges lie completely within the separated components that are of half the size.

Since the algorithm always recurse on graphs of size at least half the previous iteration, we see that $|H| \leq n/2^i$ for any $H \in CC(\mathcal{E}_{\mathcal{I}_i}(G^*))$. Thus, all chain components will become singletons after $\mathcal{O}(\log n)$ iterations and the algorithm terminates with a fully oriented graph. $\qquad\square$

**Theorem 15.** *Fix an essential graph $\mathcal{E}(G^*)$ with an unknown underlying ground truth DAG $G^*$. Given $k = 1$, Algorithm 1 runs in polynomial time and computes an atomic intervention set $\mathcal{I}$ in a deterministic and adaptive manner such that $\mathcal{E}_{\mathcal{I}}(G^*) = G^*$ and $|\mathcal{I}| \in \mathcal{O}(\log(n) \cdot \nu_1(G^*))$.*

*Proof.* Algorithm 1 runs in polynomial time because 1/2-clique separators can be computed efficiently (see Theorem 20).

Fix an arbitrary iteration $i$ of Algorithm 1 and let $G_i$ be the partially oriented graph obtained after intervening on $\mathcal{I}_i$. By Lemma 21, $\sum_{H \in CC(\mathcal{E}_{\mathcal{I}_i}(G^*))} \lfloor \frac{\omega(H)}{2} \rfloor \leq \nu_1(G^*)$. By definition of $\omega$, we always have $|K_H| \leq \omega(H)$. Thus, Algorithm 1 uses at most $2 \cdot \nu_1(G^*)$ interventions in each iteration.

By Lemma 36, there are $\mathcal{O}(\log n)$ iterations and so $\mathcal{O}(\log(n) \cdot \nu_1(G^*))$ atomic interventions are used by Algorithm 1. $\qquad\square$

**Lemma 37** (Lemma 1 of [SKDV15]). *Let $(n, k, a)$ be parameters where $k \leq n/2$. There is a polynomial time labeling scheme that produces distinct $\ell$ length labels for all elements in $[n]$ using letters from the integer alphabet $\{0\} \cup [a]$ where $\ell = \lceil \log_a n \rceil$. Further, in every digit (or position), any integer letter is used at most $\lceil n/a \rceil$ times. This labelling scheme is a separating system: for any $i, j \in [n]$, there exists some digit $d \in [\ell]$ where the labels of $i$ and $j$ differ.*

**Theorem 16.** *Fix an essential graph $\mathcal{E}(G^*)$ with an unknown underlying ground truth DAG $G^*$. Given $k > 1$, Algorithm 1 runs in polynomial time and computes a bounded size intervention set $\mathcal{I}$ in a deterministic and adaptive manner such that $\mathcal{E}_{\mathcal{I}}(G^*) = G^*$ and $|\mathcal{I}| \in \mathcal{O}(\log(n) \cdot \log(k) \cdot \nu_k(G^*))$.*

*Proof.* Algorithm 1 runs in polynomial time because 1/2-clique separators can be computed efficiently (see Theorem 20). The label computation of [SKDV15, Lemma 1] also runs in polynomial time.

Fix an arbitrary iteration $i$ of Algorithm 1 and let $G_i$ be the partially oriented graph obtained after intervening on $\mathcal{I}_i$. Since $\mathcal{E}(G_i) \subseteq \mathcal{E}(G^*)$, we see that $\nu_1(G_i) \leq \nu_1(G^*)$. Algorithm 1 intervenes on

$$|C_i| \leq \left\lceil \frac{|Q|}{k'} \right\rceil \cdot \left\lceil \log_{\lceil \frac{|Q|}{k'} \rceil} |Q| \right\rceil$$

sets of bounded size interventions, where $k' = \min\{k, |Q|/2\} > 1$.

Since $\nu_1(G^*) \geq \sum_{H \in CC(\mathcal{E}(G^*))} \lfloor \omega(H)/2 \rfloor$, we know from Lemma 30 that

$$\nu_k(G^*) \geq \left\lceil \frac{\nu_1(G^*)}{k} \right\rceil \geq \left\lceil \frac{1}{k} \cdot \sum_{H \in CC(\mathcal{E}(G^*))} \left\lfloor \frac{\omega(H)}{2} \right\rfloor \right\rceil .$$

Since $Q$ is the union of clique separator nodes, we have that $|Q| \leq \sum_{H \in CC(\mathcal{E}(G^*))} \omega(H)$ and so $\nu_k(G^*) \in \Omega(|Q|/k)$. Note that $\nu_k(G^*) \geq 1$ always.

**Case 1:** $k \leq |Q|/2$. Then, $k' = k$ and

$$|C_i| \leq \left\lceil \frac{|Q|}{k} \right\rceil \cdot \left\lceil \log_{\lceil \frac{|Q|}{k} \rceil} |Q| \right\rceil \leq \left\lceil \frac{|Q|}{k} \right\rceil \cdot \left\lceil \frac{\log |Q|}{\log \frac{|Q|}{k}} \right\rceil \leq \left\lceil \frac{|Q|}{k} \right\rceil \cdot \lceil \log(k) + 1 \rceil \in \mathcal{O}\left( \frac{|Q|}{k} \cdot \log(k) \right)$$

**Case 2:** $k \geq |Q|/2$. Then, $k' = |Q|/2$ and

$$|C_i| \leq 2 \cdot \lceil \log_2 |Q| \rceil \in \mathcal{O}(\log(k))$$

In either case, we see that $|C_i| \in \mathcal{O}\left(\nu_k(G^*) \cdot \log k\right)$. By Lemma 36, there are $\mathcal{O}(\log n)$ iterations and so $\mathcal{O}(\log(n) \cdot \log(k) \cdot \nu_k(G^*))$ bounded size interventions are used by Algorithm 1. $\qquad\square$

## G   Example illustrating the power of Lemma 21

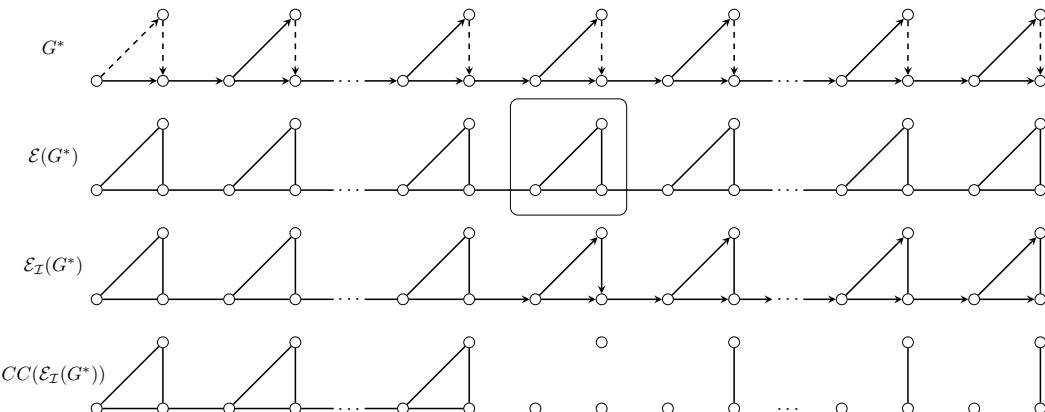

Figure 7: A DAG $G^*$ where minimum vertex cover of the unoriented covered edges (dashed arcs) is much larger than the size of the maximal clique (triangle): $\nu_1(G^*) \approx n$ while the lower bound of [SMG+20] on $\mathcal{E}(G^*)$ is a constant. Let $\mathcal{I}$ be an atomic intervention set on the middle triangle, i.e. the three vertices boxed up in $\mathcal{E}(G^*)$. The partially directed graph $\mathcal{E}_{\mathcal{I}}(G^*)$ shows the learnt arc directions after intervening on $\mathcal{I}$ and applying Meek rules. Applying the lower bound of [SMG+20] on $CC(\mathcal{E}_{\mathcal{I}}(G^*))$ now gives a much stronger lower bound of $\approx n$ due to the single edge components.

## H   Experiments and implementation

Our code and entire experimental setup is available at https://github.com/cxjdavin/verification-and-search-algorithms-for-causal-DAGs.

### H.1   Implementation details

**Verification**   We implemented our verification algorithm and tested its correctness on some well-known graphs such as cliques and trees for which we know the exact verification number.

**Search**   We use FAST CHORDAL SEPARATOR algorithm in [GRE84] to compute a chordal graph separator. This algorithm first computes a perfect elimination ordering of a given chordal graph and we use Eppstein's LexBFS implementation (https://www.ics.uci.edu/~eppstein/PADS/LexBFS.py) to compute such an ordering.

## H.2 Experiments

We base our evaluation on the experimental framework of [SMG+20] (https://github.com/csquires/dct-policy) which empirically compares atomic intervention policies. The experiments are conducted on an Ubuntu server with two AMD EPYC 7532 CPU and 256GB DDR4 RAM.

### H.2.1 Synthetic graph classes

The synthetic graphs are random connected DAGs whose essential graph is a single chain component (i.e. moral DAGs in [SMG+20]'s terminology). Below, we reproduce the synthetic graph generation procedure from [SMG+20, Section 5].

1. Erdős-Rényi styled graphs
   These graphs are parameterized by 2 parameters: $n$ and density $\rho$. Generate a random ordering $\sigma$ over $n$ vertices. Then, set the in-degree of the $n^{th}$ vertex (i.e. last vertex in the ordering) in the order to be $X_n = \max\{1, \mathtt{Binomial}(n-1, \rho)\}$, and sample $X_n$ parents uniformly form the nodes earlier in the ordering. Finally, chordalize the graph by running the elimination algorithm of [KF09] with elimination ordering equal to the reverse of $\sigma$.

2. Tree-like graphs
   These graphs are parameterized by 4 parameters: $n$, degree $d$, $e_{\min}$, and $e_{\max}$. First, generate a complete directed $d$-ary tree on $n$ nodes. Then, add $\mathtt{Uniform}(e_{\min}, e_{\max})$ edges to the tree. Finally, compute a topological order of the graph by DFS and triangulate the graph using that order.

### H.2.2 Algorithms benchmarked

The following algorithms perform *atomic interventions*. Our algorithm `separator` perform atomic interventions when given $k = 1$ and *bounded size interventions* when given $k > 1$.

`random:` A baseline algorithm that repeatedly picks a random non-dominated node (a node that is incident to some unoriented edge) from the interventional essential graph

`dct:` DCT Policy of [SMG+20]

`coloring:` Coloring of [SKDV15]

`opt_single:` OptSingle of [HB14]

`greedy_minmax:` MinmaxMEC of [HG08]

`greedy_entropy:` MinmaxEntropy of [HG08]

`separator:` Our Algorithm 1. It takes in a parameter $k$ to serve as an upper bound on the number of vertices to use in an intervention.

### H.2.3 Metrics measured

Each experiment produces 4 plots measuring "average competitive ratio", "maximum competitive ratio", "intervention count", and "time taken". For any fixed setting, 100 synthetic DAGs are generated as $G^*$ for testing, so we include error bars for "average competitive ratio", "average intervention count", and "time taken" in the plots. For all metrics, "lower is better".

The competitive ratio for an input DAG $G^*$ is measured in terms of *total atomic interventions used to orient the essential graph of $G^*$ to become $G^*$*, divided by *minimum number of interventions needed to orient $G^*$* (i.e. the verification number $\nu_1(G^*)$ of $G^*$). For non-atomic interventions, we know (Lemma 30) that $\nu_k(G^*) \geq \lceil \nu_1(G^*)/k \rceil$, so we use $\lceil \nu_1(G^*)/k \rceil$ as the denominator of the competitive ratio computation. While the competitive ratio increases as $k$ increases (Theorem 16), the number of interventions used decreases as $k$ increases.

Time is measured as the total amount of time taken to finish computing the nodes to intervene and performing the interventions. Note that our algorithm can beat `random` in terms of runtime in some cases because `random` uses significantly more interventions and hence more overall computation.

### H.2.4 Experimental results

Qualitatively, our Algorithm 1 with $k = 1$ has a similar competitive ratio to the current best-known atomic intervention policies in the literature (DCT and Coloring) while running significantly faster for some graphs (roughly ~10x faster on tree-like graphs).

**Experiment 1** Graph class 1 with $n \in \{10, 15, 20, 25\}$ and density $\rho = 0.1$. This is the same setup as [SMG$^+$20]. Additionally, we run Algorithm 1 with $k = 1$. See Fig. 8.

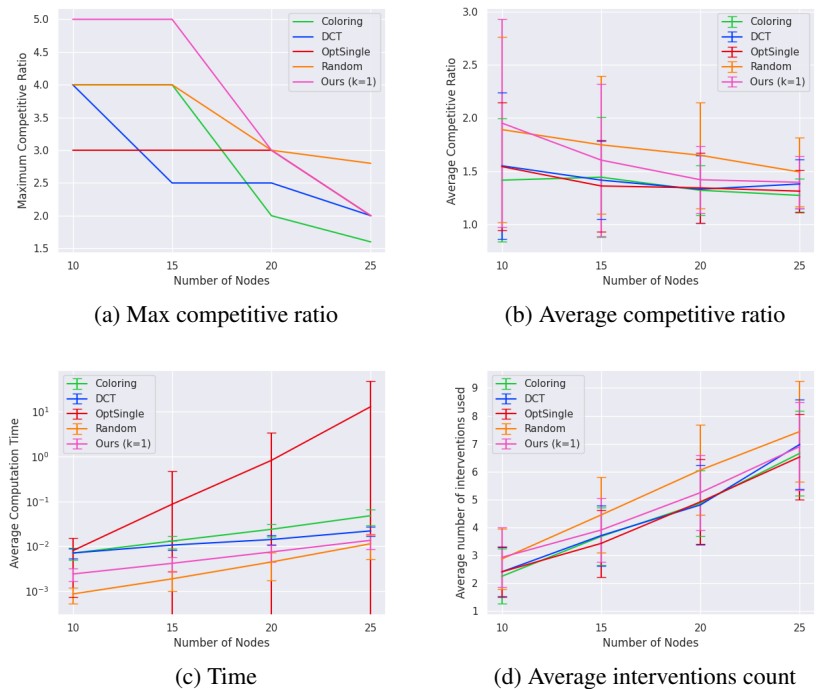

(a) Max competitive ratio

(b) Average competitive ratio

(c) Time

(d) Average interventions count

Figure 8: Plots for experiment 1

**Experiment 2** Graph class 1 with $n \in \{8, 10, 12, 14\}$ and density $\rho = 0.1$. This is the same setup as [SMG$^+$20]. Additionally, we run Algorithm 1 with $k = 1$. Note that this is the same graph class as experiment 1, but on smaller graphs because some slower algorithms are being run. See Fig. 9.

**Experiment 3** Graph class 2 with $n \in \{100, 200, 300, 400, 500\}$ and $(\text{degree}, e_{\min}, e_{\max}) = (4, 2, 5)$. This is the same setup as [SMG$^+$20]. Additionally, we run Algorithm 1 with $k = 1$. See Fig. 10.

**Experiment 4** Graph class 1 with $n \in \{10, 15, 20, 25\}$ and density $\rho = 0.1$. We run Algorithm 1 with $k \in 1, 2, 3, 5$ on the same graph class as experiment 1, but on larger graphs. See Fig. 11.

**Experiment 5** Graph class 2 with $n \in \{100, 200, 300, 400, 500\}$ and $(\text{degree}, e_{\min}, e_{\max}) = (40, 20, 50)$. We run Algorithm 1 with $k \in \{1, 2, 3, 5\}$ on the same graph class as experiment 3, but on denser graphs. See Fig. 12.

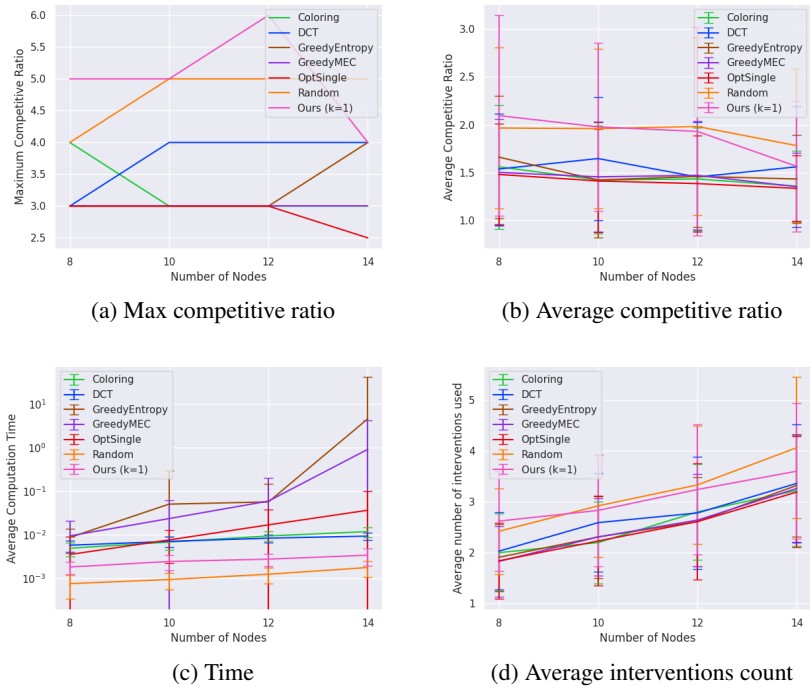

(a) Max competitive ratio

(b) Average competitive ratio

(c) Time

(d) Average interventions count

Figure 9: Plots for experiment 2

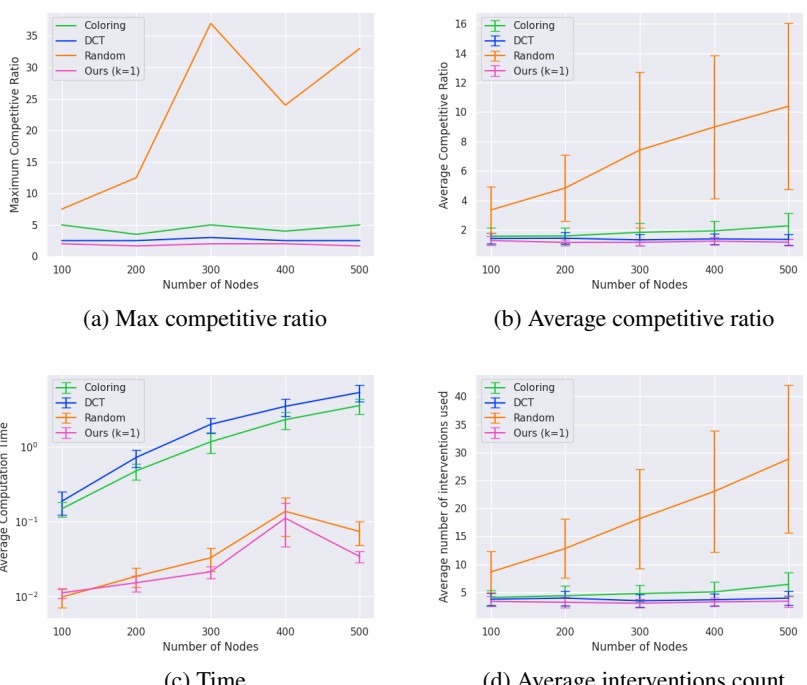

(a) Max competitive ratio

(b) Average competitive ratio

(c) Time

(d) Average interventions count

Figure 10: Plots for experiment 3

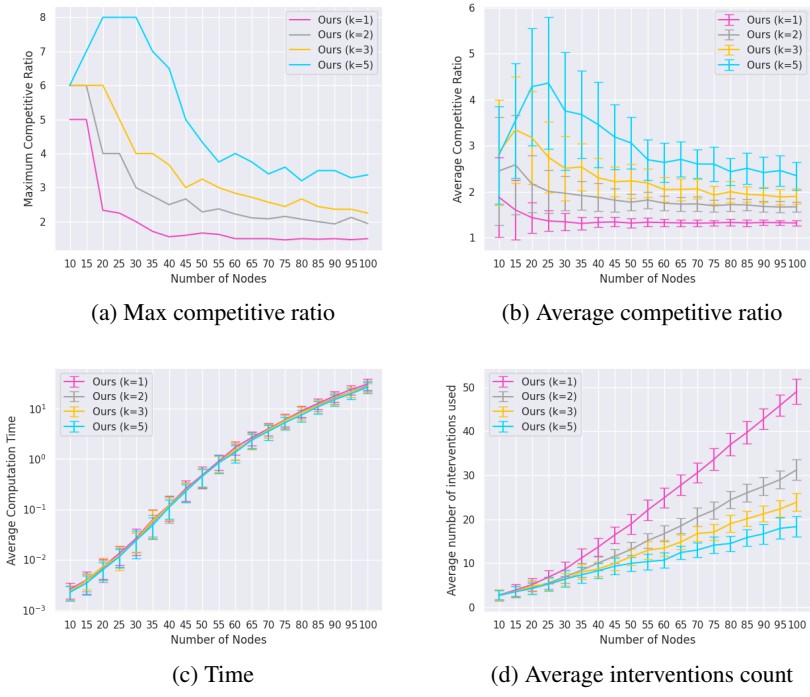

(a) Max competitive ratio

(b) Average competitive ratio

(c) Time

(d) Average interventions count

Figure 11: Plots for experiment 4

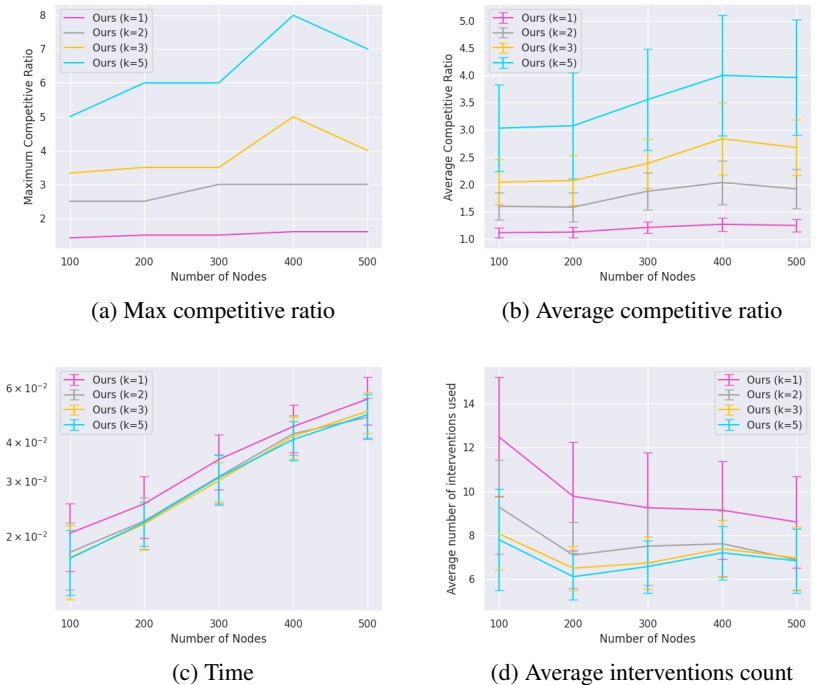

(a) Max competitive ratio

(b) Average competitive ratio

(c) Time

(d) Average interventions count

Figure 12: Plots for experiment 5