# OpenReview forum: "Verification and search algorithms for causal DAGs"
_NeurIPS.cc/2022/Conference — NeurIPS 2022 Accept_

### Official Review · Reviewer_A2Kf · 2022-07-11

**Rating:** 5
**Confidence:** 4
**Soundness:** 3 good
**Presentation:** 3 good
**Contribution:** 2 fair

**Summary:**

This paper considers the problem of recovering causal graphs from interventional data (search problem). Additionally to recovering of an unknown causal graph, the authors studied the problem of verifying whether a given directed causal graph is indeed the correct one using an optimal number of interventions (verification problem). They proposed lower bounds for both of the considered questions which are based on the minimum verification number. Finally, they proposed an algorithm for both problems and derived bounds on the number of performed interventions.

**Questions:**

- Lemma 35 looks different from Lemma 6 in [SMG+20], but in the paper, it is asserted that they are the same. How can Lemma 35 be obtained from Lemma 6 [SMG+20]?
- Why the proposed search algorithm is better than already existing algorithms and what are the main differences between them?


**Limitations:**

The authors briefly mentioned possible limitations for transferring the results to the real world, which are reasonable. All the limitations do not have a negative social impact.


**Strengths And Weaknesses:**

Strength:
- The authors studied an interesting connection between the optimal number of interventions that are needed to verify whether the given directed causal graph is correct and the optimal number of interventions that one needs to recover the causal graph. Furthermore, they compare their results with existing works.

Weaknesses:
- Despite there is an example that the minimum verification number is larger than the lower bounds derived in [PSS22], it is not clear how much this bound is better. It seems that the difference between these two values may be just a constant or just a constant ratio.
- Lemma 35 is different from the lemma introduced in [SMG+20] and it is not obvious how it is possible to get Lemma 35 from Lemma 6 in [SMG+20]. Lemma 35 is then used for the proof of Lemma 21, which are one of the main results in comparing achieved lower bounds with another work.
- The proposed algorithm is fully composed of steps irrelevant to the minimum verification number. Moreover, the whole analysis of the algorithm follows the theorems from the other works, and it is not unclear how this algorithm is different from already existent ones and why it would have better performance.
- No experiments were done for comparing the proposed algorithm with already existing algorithms.
- Some of the results about minimum verification number are easy corollaries of already existing theorems (such as Theorem 11 and 12).

---

> ### Author Response · Authors · 2022-08-02
> **Author Response to Reviewer A2Kf (Part 1/2)**
>
> We thank the reviewer for their time and valuable feedback. In the following, we provide our responses to the concerns/questions raised in the review.
>
> ## Significance of windmill graph example
>
> It is no surprise that the gap between the verification number of the windmill graph and the lower bound given by [PSS22] is small. In fact, [PSS22] already gives an *almost tight characterization* of the verification number by proving that the ratio is at most a factor of 2 (they also ran experiments); meanwhile, we present an *exact characterization* of the verification number that is easily computable in polynomial time in our work. The goal of the windmill graph construction was to provide a small explicit graph example highlighting the gap in the understanding of the verification number while showcasing the usefulness of our covered edges perspective. While constant ratios may not seem much, having an exact simple characterization greatly helps in advancing the understanding of causal discovery.
>
> ## Search algorithm having steps irrelevant to minimum verification number
>
> Verification and search are two fundamentally different problems. For verification, the ground truth DAG $G^*$ is given. For search, only the essential graph is given as input and $G^*$ is *not* given. Although verification is irrelevant to our algorithm for the search problem, we remark that both these problems are important and interesting in their own right. In practice, we are often only given the essential graph (obtained via observational data) and would only be able to perform a search. Meanwhile, understanding the verification problem allows us to measure ``how much easier the search problem would have been if we knew the ground truth''. By definition, the verification number is the minimum possible interventions needed and hence it serves as a natural benchmark to measure search algorithms against.
>
> Prior to our work, all previous works either used lower bound or exponential brute force algorithms to compute the verification number when computing the benchmark to compare their search algorithms against. Therefore, we believe that the exact characterization of the verification number via a polynomial time algorithm not only furthers our understanding but also has some immediate applications such as allowing us to benchmarking search algorithms on even larger graphs. What we really find fascinating is that we can design search algorithms (Algorithm 1) with provable guarantees that is competitive (up to logarithmic factors) against the verification number $\nu(G^*)$, despite not being aware what $\nu(G^*)$ is.
>
> ## Difference of our algorithm compared to prior works
>
> Our algorithm is different from existing algorithms both algorithmically and in terms of its theoretical guarantees (see the discussion on lines 293-298). While our analysis reuses some theorems from existing work, these existing results were insufficient on their own. Crucially, we need Lemma 21 (a strengthened lower bound of [SMG+20] that we proved) in order to obtain the theoretical guarantee of our search algorithm. Prior to our work, the best known provable guarantees for search algorithms were by [SMG+20] but their guarantees only hold for special graphs. Meanwhile, our algorithm has the same provable guarantees but these guarantees hold for any general graphs. Please let us know if we missed any references that you think closely resembles our work.
>
> ## Some results are easy corollaries
>
> We think that the statements of our results are interesting in their own right and proving them without the framework used in our paper is non-trivial. The reason why some of our proofs appear simple is because of our improved understanding of the causal discovery problem (via our covered edges perspective) and the structure of our proofs (for example, some of our induction arguments are subtle to design but easy to follow). We believe that our covered edge characterization of the verification problem is one of the strengths of our paper as it advances our understanding of causal discovery while enabling simpler proofs.
>
> ## References
>
> [PSS22] Vibhor Porwal, Piyush Srivastava, and Gaurav Sinha. Almost Optimal Universal Lower Bound for Learning Causal DAGs with Atomic Interventions. In The 25th International Conference on Artificial Intelligence and Statistics, 2022.
>
> [SMG+20] Chandler Squires, Sara Magliacane, Kristjan Greenewald, Dmitriy Katz, Murat Kocaoglu, and Karthikeyan Shanmugam. Active Structure Learning of Causal DAGs via Directed Clique Trees. Advances in Neural Information Processing Systems, 33:21500–21511, 2020

---

> ### Author Response · Authors · 2022-08-02
> **Author Response to Reviewer A2Kf (Part 2/2)**
>
> ## Our Lemma 35 versus Lemma 6 of [SMG+20]
>
> Thank you for bringing this to our attention.  For immediate access, we first restate the results from both papers below.
>
> Our Lemma 35:
> > Let $G$ be a moral DAG. Then, $\nu_1(G) \geq \lfloor \frac{\omega(skeletonsyd(G))}{2} \rfloor$.
>
> Lemma 6 of [SMG+20]:
> > Let $D$ be a moral DAG and let $G$ = skel($D$). Then $m(D) \geq \lfloor \frac{\omega(G)}{2} \rfloor$, where $\omega(G)$ is the size of the largest clique in G.
>
> Note that the word "skeletonsyd" in Lemma 35 is a typo and should be "skeleton", which is exactly the same as "skel" in [SMG+20] (in the preliminaries, they write "The skeleton of graph $D$, skel($D$), is the undirected graph with the same vertices and adjacencies as $D$"). In the revision, we will fix this typo.
>
> In addition, note that Definition 7 of [SMG+20] writes,
> > Given a general DAG $D$, a verifying intervention set (VIS) is a set of single-node interventions $\mathcal{I}$ that fully orients the DAG starting from an essential graph $\ldots$ We denote the size of the minimal VIS for $D$ as $m(D)$.
>
> That is, $m(D)$ corresponds to our notation of $v_1(G)$. We have also defined $\omega(G)$ in the preliminaries section of our main paper. Therefore, up to notation differences, these statements are the same. Please let us know if we misunderstood your question.
>
> ## References
>
> [SMG+20] Chandler Squires, Sara Magliacane, Kristjan Greenewald, Dmitriy Katz, Murat Kocaoglu, and Karthikeyan Shanmugam. Active Structure Learning of Causal DAGs via Directed Clique Trees. Advances in Neural Information Processing Systems, 33:21500–21511, 2020

---

> ### Author Response · Authors · 2022-08-02
> **Empirical evaluation**
>
> We thank the suggestion of some reviewers to empirically evaluate our algorithms. In response, we have coded up our algorithms and ran some experiments. We will supplement a more extensive experimental evaluation in a new appendix section in our paper revision. As we cannot upload images in these text boxes, we provide them in an anonymous Google Drive: https://drive.google.com/drive/folders/1QNZR7j73zGnHBSMBzyzB_YNb09XfMpiJ?usp=sharing. In this folder, we also provide instructions on how to reproduce our experiments and more elaboration on the plots generated from the experiments.
>
> Despite the favourable empirical outcomes from these experiments, we wish to emphasize that we still believe that the main contribution of our work is a theoretical understanding and theoretically provable algorithms for the verification and search problems. What we really find fascinating is that we can design search algorithms (Algorithm 1) with provable guarantees that is competitive (up to logarithmic factors) against the verification number $\nu(G^*)$, despite not being aware what $\nu(G^*)$ is.
>
> ### Verification experiments
>
> Before we talk about the experiments, we want to emphasize that verification is a basic but important problem. For instance, an efficient algorithm for computing exact verification numbers is important for benchmarking search algorithms. Prior works that ran experiments for search algorithms had to either use a lower bound for the verification number or compute it via exponential brute force search, which is impractical for large graph sizes.
>
> The verification experiments ran by [PSS22] were to validate that their lower bound is within a factor of 2 of the true verification number, and also empirically compare their lower bound against the lower bound of [SMG+20]. As we have an exact characterization of the verification number and a practical efficient algorithm to compute it exactly, we believe that running similar experiments would not be fruitful. Instead, we have coded up our verification algorithm and tested its correctness on some well-known graphs such as cliques and trees for which we know the exact verification number. We provide the source file in our anonymous folder.
>
> ### Search experiments
>
> We ran search experiments in the framework of [SMG+20] (https://github.com/csquires/dct-policy) which empirically compares atomic intervention policies -- see Section 5 ``Experimental Results'' in [SMG+20] for a description and details of the simulated graphs. We implemented two variants of our search algorithm (Algorithm 1 with $k=1$) where we either deterministically intervene on all nodes in our clique separator one at a time, or randomly pick one until all edges are incident to the clique separator have been oriented. We implemented two variants because it is known [Ebe10] that random atomic interventions on cliques are expected to perform better than deterministic atomic interventions by a constant multiplicative factor, and we thought that it would be interesting to compare these variants empirically. Qualitatively, on large graphs, our proposed algorithm has a similar empirical performance to the current best-known atomic intervention policies in the literature (`DCT` and `Coloring`) in terms of the competitive ratio (i.e. the number of interventions used divided by the verification number of each graph) while running significantly faster (roughly 10x faster). In addition, note that our algorithm has provable guarantees that it has a $O(\log n)$ competitive ratio with respect to the verification number for any general graph. The previously best-known theoretical guarantee for this problem is by [SMG+20]: they also show an $O(\log n)$ approximation but their proofs only hold for special graph classes. In the revision, we will run our algorithm for other graphs and also include the implementation for non-atomic interventions.
>
> ### Implementation details of our search algorithm
>
> Our implementation of the chordal graph separator is the `FAST CHORDAL SEPARATOR` algorithm in [GRE84] which first computes a perfect elimination ordering of a given chordal graph. To do so, we use Eppstein's LexBFS implementation (https://www.ics.uci.edu/~eppstein/PADS/LexBFS.py).
>
> ### Reproducibility
>
> For reproducibility purposes of our search experiments, we provide the exact modifications made and a bash script that anyone can run to directly download all the necessary files, modifies the `dct-policy` codebase, run the experiments, and generate these plots automatically. We also provide an implementation of our verification algorithm in a Python script. See the README in the Google Drive folder for details.
>
> ## References
>
> [Ebe10] Frederick Eberhardt. Causal Discovery as a Game. In Causality: Objectives and Assessment, pages 87–96. PMLR, 2010
>
> [GRE84] John R. Gilbert, Donald J. Rose, and Anders Edenbrandt. A Separator Theorem for Chordal Graphs. SIAM Journal on Algebraic Discrete Methods, 5(3):306–313, 1984

---

> ### Comment · Reviewer_A2Kf · 2022-08-08
> **Response to rebuttal**
>
> Thank you for your thorough and detailed answers.
>
> Lemma 35 and Lemma 6 of [SMG+20] are indeed the same. The most significant result of this paper is realizing and proving that Theorem 9 (criterion for the set to be minimum verifying set) follows from Lemma 7 [Chi 95]. Theorem 9 is indeed a new and interesting result that allows authors to compute $v_1(G)$ exactly and efficiently. However other results are mostly straightforward corollaries of previous works. Unfortunately, I still believe that this work lacks novelty. For example, the idea of considering a minimal verifying set is well studied in [SMG+20], and the main results in this work are similar to ones in [SMG+20] with slight modifications/extensions (for instance, Lemma 21 and Theorem 2 [SMG +20]).

---

> > ### Author Response · Authors · 2022-08-09
> > **Response to reviewer A2Kf**
> >
> > Thank you for your response.
> >
> > Regarding your first comment, we want to remark that our proof for Theorem 9 (see Appendix E) does **not** refer to Lemma 7 in any way and we would really appreciate if you would let us know how (our) Theorem 9 follows from Lemma 7 [Chi 95].
> >
> > Regarding your second comment, note that our Lemma 21 may look similar to Theorem 2 of [SMG+20] but our result is **stronger**: Appendix G gives an illustration of how it can be stronger by even a linear factor in the size of the graph. Also, the upper bound guarantees of [SMG+20] only hold for intersection-comparable graphs while ours hold for **any general graph**; In fact, our search algorithm is much simpler and different from all the previous approaches. Independent of the techniques used in our proofs (which may look simple in hindsight), we believe that the guarantees of our search algorithm are very interesting as they hold for **any arbitrary graphs** and to the best of our knowledge remained an **open problem** until our work.
> >
> > Furthermore, our empirical experiments (see our previous rebuttal response) show that our search algorithm is comparable to the state of the art (in terms of number of interventions used) while running around **10x faster** on larger graphs.
> >
> > Thank you for your time and patience.

---

> > > ### Comment · Reviewer_A2Kf · 2022-08-09
> > > **Thanks for the clarification**
> > >
> > > Thanks for addressing the comments.
> > >
> > > > “Regarding your first comment, we want to remark that our proof for Theorem 9 (see Appendix E) does not refer to Lemma 7 in any way and we would really appreciate if you would let us know how (our) Theorem 9 follows from Lemma 7 [Chi 95]”
> > >
> > > Yes, the proof does not refer to Lemma 7, but it seems that Theorem 9 follows from Lemma 7 in the following way:
> > >
> > > Necessary:  If one does not orient all covered edges (which are undirected in the essential graph), then any of them can be flipped and get another equivalent graph according to [Chi 95].
> > >
> > > Sufficiency: The set of covered edges with their orientations determines the graph uniquely (follows from [Chi 95]), that is if one orients them then the graph is identified.
> > >
> > > >“Lemma 21 may look similar to Theorem 2 of [SMG+20] but our result is stronger”
> > >
> > > Lemma 21 is stronger because instead of using Theorem 2 for one DAG, a wider set of DAGs (which comes from interventions on the original graph)  was considered, and then the best graph using the same bound from Theorem 2 (maximization over I-essential graphs) was selected.
> > >
> > > >“Also, the upper bound guarantees of [SMG+20] only hold for intersection-comparable graphs while ours hold for any general graph”
> > >
> > > I partly disagree with the authors. For example, in [SMG+20], it is asserted:
> > > “The size of the problem is cut in half after each clique-intervention, so that we use at most P G∈CC(E(D))dlog2 (|C(G)|) clique-interventions, where C(G) is the set of maximal cliques for G.” that refer to an arbitrary graph G and which somehow similar to the idea of the proposed algorithm here. However, I agree that they did not propose an upper bound on the algorithm for any arbitrary graph. Moreover, it is interesting to see that the performance of the algorithm is better in terms of time complexity with comparable results to the state of the art.
> > >
> > > Based on the above discussions, I decided to change my score from 4 to 5.

---

> > > > ### Author Response · Authors · 2022-08-09
> > > > **Response to reviewer A2Kf**
> > > >
> > > > Thank you so much for your time. *We really appreciate your responses and are glad that we could have this discussion!* Please refer to the following for our responses.
> > > >
> > > > ### Comment 1: Theorem 9 via Lemma 7
> > > >
> > > > On necessity: While your statement is true, it is insufficient as a proof for the necessity. For example, why is it impossible for someone to orient all the covered edges by intervening on some other vertices and then applying Meek rules? To this end, one has to argue that it is *impossible* to orient a covered edge unless some intervention explicitly intervenes one of the endpoints of that covered edge.
> > > >
> > > > On sufficiency: We agree that sufficiency follows from the statement "The set of covered edges with their orientations determines the graph uniquely". However, we are not aware of such a result and thus had to prove that non-covered edges will also be oriented. Could you kindly point us to the specific lemma/theorem that we have missed?
> > > >
> > > > ### Comment 2: Lemma 21
> > > >
> > > > Thank you for appreciating Lemma 21. We would like to highlight that the additional "max" is an important observation and it allows one to obtain simple analysis of competitive bounds by arranging/grouping the interventions used appropriately.
> > > >
> > > > ### Comment 3: Our search algorithm and analysis is not novel enough
> > > >
> > > > While some of the techniques used in our work are standard in algorithm design, we would like to highlight that our novelty is in the usage of graph separators to enable us to bypass the need to assume intersection-incomparability in our analysis.

---

### Official Review · Reviewer_vQyV · 2022-07-11

**Rating:** 7
**Confidence:** 2
**Soundness:** 4 excellent
**Presentation:** 3 good
**Contribution:** 3 good

**Summary:**

The authors study the problems of verification and search for identifying causal DAGs from minimal-cost interventions. Using the concept of covered edges, they provide a theoretical result characterizing verifying sets, as well as a practical method for constructing such sets optimally. For the problem of search, the paper provides a novel algorithm for search, with complexity bounded by a connection to the minimum cost verifying set.

**Questions:**

1) Have the authors considered adaptive solutions for the verification problem also, and whether this might provide further benefit?


**Limitations:**

Yes, limitations and social impact have been addressed explicitly in the text.

**Strengths And Weaknesses:**

Overall, this is a strong theoretical work which provides novel perspectives on minimal-cost identification of causal DAGs. The authors make a number of novel connections between the verification and search problems, some of which in particular are very interesting and even surprising:
- Thm 9, which states that it is N+S to separate all covered edges in G to verify G (reduce G’s I-MEC to a singleton); this is a stronger version of prior results on the search problem (e.g. Thm 1 in [KDV17]) which say it is N+S to separate all edges to find the true graph (reduce all I-MECs to singletons).
- That the search problem can be algorithmically linked to the verification problem by Lemma 21, with a strategy of intervening on cliques whose size can be bounded by intervention sizes for the verification problem.

The theoretical connections also give rise to efficient algorithms for performing verification and search on general graphs, which are a significant advance over prior approaches.

From a presentation standpoint, the paper is generally well written with clear presentation of the results and comparison to prior work. However, it is also very dense, particularly with the novel results and algorithms, which are packed into the final 4 pages of the paper. For instance, Algorithm 1 would benefit from explanation of the function/intuition behind each step, as well as bounding the complexity. I would suggest that the authors move some of the (arguably) more peripheral content to the Appendix (e.g. lines 258-277), to make more space to explain the key results.

---

> ### Author Response · Authors · 2022-08-02
> **Author Response to Reviewer vQyV**
>
> We thank the reviewer for their time and valuable feedback. In the following, we provide our responses to the concerns/questions raised in the review.
>
> ## Adaptive solutions to verification
>
> As our verification result is computed in a non-adaptive manner, it is indeed natural to wonder if adaptive solutions to the verification problem can be even better. However, note that it could be the case that the given advice graph $G$ is indeed $G^*$ (i.e. any revealed arc orientations from interventions agrees with $G$). Thus, in the worst case, adaptivity will not bring any benefit to solving the verification problem.
>
> ## Perceived denseness of write-up due to space constraints
>
> As this paper covers two fundamental problems, we needed to introduce several definitions and notations. We have put an effort to remove non-essential text and pushed some of the prior work and formal proofs into the appendix while focusing on motivating our results and giving the high-level intuition of our proof techniques in the main paper. In our revision, we will try to improve the presentation further. Regarding lines 258-277, they serve to motivate why one would be interested in the weighted verification problem. Since the weighted verification cost is the least possible cost incurred by any interventional strategy in order to fully orient an essential graph (assuming the ground truth DAG is known), it is one of the key benchmarks to compare against weighted search algorithms (which do not know the ground truth DAG).

---

> ### Author Response · Authors · 2022-08-02
> **Empirical evaluation**
>
> We thank the suggestion of some reviewers to empirically evaluate our algorithms. In response, we have coded up our algorithms and ran some experiments. We will supplement a more extensive experimental evaluation in a new appendix section in our paper revision. As we cannot upload images in these text boxes, we provide them in an anonymous Google Drive: https://drive.google.com/drive/folders/1QNZR7j73zGnHBSMBzyzB_YNb09XfMpiJ?usp=sharing. In this folder, we also provide instructions on how to reproduce our experiments and more elaboration on the plots generated from the experiments.
>
> Despite the favourable empirical outcomes from these experiments, we wish to emphasize that we still believe that the main contribution of our work is a theoretical understanding and theoretically provable algorithms for the verification and search problems. What we really find fascinating is that we can design search algorithms (Algorithm 1) with provable guarantees that is competitive (up to logarithmic factors) against the verification number $\nu(G^*)$, despite not being aware what $\nu(G^*)$ is.
>
> ### Verification experiments
>
> Before we talk about the experiments, we want to emphasize that verification is a basic but important problem. For instance, an efficient algorithm for computing exact verification numbers is important for benchmarking search algorithms. Prior works that ran experiments for search algorithms had to either use a lower bound for the verification number or compute it via exponential brute force search, which is impractical for large graph sizes.
>
> The verification experiments ran by [PSS22] were to validate that their lower bound is within a factor of 2 of the true verification number, and also empirically compare their lower bound against the lower bound of [SMG+20]. As we have an exact characterization of the verification number and a practical efficient algorithm to compute it exactly, we believe that running similar experiments would not be fruitful. Instead, we have coded up our verification algorithm and tested its correctness on some well-known graphs such as cliques and trees for which we know the exact verification number. We provide the source file in our anonymous folder.
>
> ### Search experiments
>
> We ran search experiments in the framework of [SMG+20] (https://github.com/csquires/dct-policy) which empirically compares atomic intervention policies -- see Section 5 ``Experimental Results'' in [SMG+20] for a description and details of the simulated graphs. We implemented two variants of our search algorithm (Algorithm 1 with $k=1$) where we either deterministically intervene on all nodes in our clique separator one at a time, or randomly pick one until all edges are incident to the clique separator have been oriented. We implemented two variants because it is known [Ebe10] that random atomic interventions on cliques are expected to perform better than deterministic atomic interventions by a constant multiplicative factor, and we thought that it would be interesting to compare these variants empirically. Qualitatively, on large graphs, our proposed algorithm has a similar empirical performance to the current best-known atomic intervention policies in the literature (`DCT` and `Coloring`) in terms of the competitive ratio (i.e. the number of interventions used divided by the verification number of each graph) while running significantly faster (roughly 10x faster). In addition, note that our algorithm has provable guarantees that it has a $O(\log n)$ competitive ratio with respect to the verification number for any general graph. The previously best-known theoretical guarantee for this problem is by [SMG+20]: they also show an $O(\log n)$ approximation but their proofs only hold for special graph classes. In the revision, we will run our algorithm for other graphs and also include the implementation for non-atomic interventions.
>
> ### Implementation details of our search algorithm
>
> Our implementation of the chordal graph separator is the `FAST CHORDAL SEPARATOR` algorithm in [GRE84] which first computes a perfect elimination ordering of a given chordal graph. To do so, we use Eppstein's LexBFS implementation (https://www.ics.uci.edu/~eppstein/PADS/LexBFS.py).
>
> ### Reproducibility
>
> For reproducibility purposes of our search experiments, we provide the exact modifications made and a bash script that anyone can run to directly download all the necessary files, modifies the `dct-policy` codebase, run the experiments, and generate these plots automatically. We also provide an implementation of our verification algorithm in a Python script. See the README in the Google Drive folder for details.
>
> ## References
>
> [Ebe10] Frederick Eberhardt. Causal Discovery as a Game. In Causality: Objectives and Assessment, pages 87–96. PMLR, 2010
>
> [GRE84] John R. Gilbert, Donald J. Rose, and Anders Edenbrandt. A Separator Theorem for Chordal Graphs. SIAM Journal on Algebraic Discrete Methods, 5(3):306–313, 1984

---

### Official Review · Reviewer_ua5U · 2022-07-17

**Rating:** 5
**Confidence:** 2
**Soundness:** 2 fair
**Presentation:** 3 good
**Contribution:** 2 fair

**Summary:**

The paper proposed recovering and verification of causal graphs via sufficient hard interventions with unlimited data.  Contributions are constructing bounds on the “minimum verification number” of a graph G belonging to the Markov equivalent class of the underlying ground truth.  Compared to existing approaches, their method could deal with general graph.  They characterize the verifying intervention sets via separation of un-oriented cover edges of the input essential graph.

They also consider a relative new setting, i.e. verifying if an expert provided causal graph is correct using minimal interventions.

Furthermore, they use an objective function to measure the optimality of intervention policies by trading  off the additive weights of verification inventions as well as the size of the verification set.
The main contribution of this paper seems to theoretical, by proving a strengthened version of previous work.

The paper is well written with clean explanations, the contributions are summarized clearly, lemmas correspondence to existing works are well indicated.

There is no conclusion section, for page limit, it looks like algorithms could still be moved to appendix to save some space for another summary at the end if the authors did not find this unnecessary.

**Questions:**

Q1. I believe a tighter bound would give better guidance in practice, before verification/adaptive search, for the experiment carrier to evaluate how many resources one might need to allocate to finish the experiment, i.e. identify the causal graph.

What would be the future work of this work, consider if this paper get cited in the future, what would be the improvements based on this paper?

Q2. If I did not miss it, there seems to be no empirical study to support the claims made in the paper, compared to SMG 20, PSS 22.
Regarding the new scheme used in this contribution, does the author think it might be helpful to carry out empirical studies to validate those bounds or show the superiority of the new policy?

Q3. In line 351, what does “arefi9x” mean?
Q4. it seems that there is no conclusion of the paper?



**Limitations:**

L1:
I would suggest conducting an empirical study validating the proposed algorithm alongside the theorems.

**Strengths And Weaknesses:**

Strengths:




Strengths:
S1. They extend previous results to general graphs and provide tighter bounds
S2. They also consider a relative new setting, i.e. verifying if an expert provided causal graph is correct using minimal interventions.
S3. They derived richer results  beyond single vertex intervention.



Weakness:

W1. The paper assume causal sufficiency, which is usually not true in reality.
As noted in the introduction part, the hard intervention is also not a realistic assumption.

W2. If I did not miss it, there seems to be no empirical study to support the claims made in the paper, compared to SMG 20, PSS 22.

---

> ### Author Response · Authors · 2022-08-02
> **Author Response to Reviewer ua5U**
>
> Author Response to Reviewer ua5U
>
> We thank the reviewer for their time and valuable feedback. In the following, we provide our responses to the concerns/questions raised in the review.
>
> ## Causal sufficiency assumption
>
> Thank you for raising this concern and we agree that the causal sufficiency assumption might not hold in reality. Despite this, the problem of causal discovery is still well-studied under the assumptions of causal sufficiency (see the extensive list of references given in our related work). Furthermore, much of its theoretical understanding still remains unknown. Our work is the first to give a complete characterization of the verification problem (see our response regarding verification experiments about why this is an important contribution to the field) and the first to give an interventional search algorithm with provable guarantees for general graphs (prior works such as [SMG+20] only had guarantees for special graph classes). We view our work as an important step (but not an end goal) towards a principled theory of causal discovery that provides theoretical guarantees while using as few assumptions as possible.
>
> ## Future work
>
> Thank you for asking this question. There are indeed many possible future directions that we think are important.
> For instance, can we work with soft interventions and design algorithms with interventional sample complexities in mind (e.g. see [KJSB19] and [ABDK18] respectively)? It is also of great interest to remove/weaken the assumptions on causal sufficiency while maintaining strong theoretical guarantees. Note that we have already included some of these discussions in the "Societal impact and limitations" section of our paper (Page 2). In our revision, we can try to make this more explicit.
>
> ## ``arefi9x'' on line 351
>
> Thank you for pointing out this typo. We will fix it in the revision. The sentence should read: Existence and efficient computation of graph separators are well studied [LT79, GHT84, GRE84, AST90, KR10, WN11] and **are** commonly used in divide-and-conquer graph algorithms and as analysis tools.
>
> ## Lack of conclusion section and suggestion to put the algorithm in the appendix to make space for it
>
> Thank you for your suggestion. We thought it would be more engaging to make relevant discussion points right after presenting each of our results. In the revision, we will add a conclusion section summarizing our results and also stating the possible future directions (discussed above) of our work. Regarding our algorithm description, we made a conscious choice to put it in the main text because we wanted to highlight its simplicity and it compliments the high-level proof outline which we presented in the text surrounding it.
>
> ## References
>
> [PSS22] Vibhor Porwal, Piyush Srivastava, and Gaurav Sinha. Almost Optimal Universal Lower Bound for Learning Causal DAGs with Atomic Interventions. In The 25th International Conference on Artificial Intelligence and Statistics, 2022
>
> [SMG+20] Chandler Squires, Sara Magliacane, Kristjan Greenewald, Dmitriy Katz, Murat Kocaoglu, and Karthikeyan Shanmugam. Active Structure Learning of Causal DAGs via Directed Clique Trees. Advances in Neural Information Processing Systems, 33:21500–21511, 2020
>
> [KJSB19] Murat Kocaoglu, Amin Jaber, Karthikeyan Shanmugam, and Elias Bareinboim. Characterization and learning of causal graphs with latent variables from soft interventions. Advances in Neural Information Processing Systems, 32, 2019
>
> [ABDK18] Jayadev Acharya, Arnab Bhattacharyya, Constantinos Daskalakis, and Saravanan Kandasamy. Learning and testing causal models with interventions. Advances in Neural Information Processing Systems, 31, 2018

---

> ### Author Response · Authors · 2022-08-02
> **Empirical evaluation**
>
> We thank the suggestion of some reviewers to empirically evaluate our algorithms. In response, we have coded up our algorithms and ran some experiments. We will supplement a more extensive experimental evaluation in a new appendix section in our paper revision. As we cannot upload images in these text boxes, we provide them in an anonymous Google Drive: https://drive.google.com/drive/folders/1QNZR7j73zGnHBSMBzyzB_YNb09XfMpiJ?usp=sharing. In this folder, we also provide instructions on how to reproduce our experiments and more elaboration on the plots generated from the experiments.
>
> Despite the favourable empirical outcomes from these experiments, we wish to emphasize that we still believe that the main contribution of our work is a theoretical understanding and theoretically provable algorithms for the verification and search problems. What we really find fascinating is that we can design search algorithms (Algorithm 1) with provable guarantees that is competitive (up to logarithmic factors) against the verification number $\nu(G^*)$, despite not being aware what $\nu(G^*)$ is.
>
> ### Verification experiments
>
> Before we talk about the experiments, we want to emphasize that verification is a basic but important problem. For instance, an efficient algorithm for computing exact verification numbers is important for benchmarking search algorithms. Prior works that ran experiments for search algorithms had to either use a lower bound for the verification number or compute it via exponential brute force search, which is impractical for large graph sizes.
>
> The verification experiments ran by [PSS22] were to validate that their lower bound is within a factor of 2 of the true verification number, and also empirically compare their lower bound against the lower bound of [SMG+20]. As we have an exact characterization of the verification number and a practical efficient algorithm to compute it exactly, we believe that running similar experiments would not be fruitful. Instead, we have coded up our verification algorithm and tested its correctness on some well-known graphs such as cliques and trees for which we know the exact verification number. We provide the source file in our anonymous folder.
>
> ### Search experiments
>
> We ran search experiments in the framework of [SMG+20] (https://github.com/csquires/dct-policy) which empirically compares atomic intervention policies -- see Section 5 ``Experimental Results'' in [SMG+20] for a description and details of the simulated graphs. We implemented two variants of our search algorithm (Algorithm 1 with $k=1$) where we either deterministically intervene on all nodes in our clique separator one at a time, or randomly pick one until all edges are incident to the clique separator have been oriented. We implemented two variants because it is known [Ebe10] that random atomic interventions on cliques are expected to perform better than deterministic atomic interventions by a constant multiplicative factor, and we thought that it would be interesting to compare these variants empirically. Qualitatively, on large graphs, our proposed algorithm has a similar empirical performance to the current best-known atomic intervention policies in the literature (`DCT` and `Coloring`) in terms of the competitive ratio (i.e. the number of interventions used divided by the verification number of each graph) while running significantly faster (roughly 10x faster). In addition, note that our algorithm has provable guarantees that it has a $O(\log n)$ competitive ratio with respect to the verification number for any general graph. The previously best-known theoretical guarantee for this problem is by [SMG+20]: they also show an $O(\log n)$ approximation but their proofs only hold for special graph classes. In the revision, we will run our algorithm for other graphs and also include the implementation for non-atomic interventions.
>
> ### Implementation details of our search algorithm
>
> Our implementation of the chordal graph separator is the `FAST CHORDAL SEPARATOR` algorithm in [GRE84] which first computes a perfect elimination ordering of a given chordal graph. To do so, we use Eppstein's LexBFS implementation (https://www.ics.uci.edu/~eppstein/PADS/LexBFS.py).
>
> ### Reproducibility
>
> For reproducibility purposes of our search experiments, we provide the exact modifications made and a bash script that anyone can run to directly download all the necessary files, modifies the `dct-policy` codebase, run the experiments, and generate these plots automatically. We also provide an implementation of our verification algorithm in a Python script. See the README in the Google Drive folder for details.
>
> ## References
>
> [Ebe10] Frederick Eberhardt. Causal Discovery as a Game. In Causality: Objectives and Assessment, pages 87–96. PMLR, 2010
>
> [GRE84] John R. Gilbert, Donald J. Rose, and Anders Edenbrandt. A Separator Theorem for Chordal Graphs. SIAM Journal on Algebraic Discrete Methods, 5(3):306–313, 1984

---

### Meta-Review · Area_Chair_jThY · 2022-08-29

**Recommendation:** Accept
**Confidence:** Certain

**Metareview:**

This paper's reviews as it stands are divergent. The scores are 7, 5 and 4. The paper has seen discussion between reviewers with negative opinion and the authors. One reviewer who engaged in discussion revised the score up by 1.

The most unfavorable reviewer's main issue was lack of empirical evaluations comparing the authors algorithms to close competitors [SMG 20, PSS 22]. - Authors have responded turning in a quick implementation with plots comparing performance of their algorithm with competitors. Authors attached it plots and a readme in the form of an anonymous Drive folder (I looked at it briefly).  It seems like their algorithm is very competitive with state of the art and infact in terms of runtime is faster than even random in some cases. Experiments seem reasonably comprehensive.  I would have ideally liked the authors to include the contents of the drive folder into the main paper and uploaded a revision (I am not sure if authors are aware that one can update the paper during the rebuttal).
I would consider this issue sort of taken care of. Empirical simulations do clearly show that the proposed algorithms are effective for random graphs of different size.

Other concerns (even after a long discussion with reviewers) are: How significant are the theoretical results in comparison to [SMG 20, PSS 22].
1) Does it follow from many theorems about covered edges [Chi 95] classically known and other theorems from these two recent references ?
Authors responded saying - they are the first to give an *exact* algorithm to perform adaptive interventions to verify if a given graph is indeed the true one exactly characterizing the instance optimal number of interventions. I agree with the authors that this is not known and relation to covered edges does not directly follow from existing classical results (as authors have explained and I did see the proofs in the supplement.)
So results for exact instance optimal verification are certainly new and novel and previous works only provided bounds on the verification number.

2) How novel are the search results ?  - Here, it is true that for proving approximation guarantee they do rely on a slight modification of a lower bound, i.e. Lemma 21 in the paper as observed by one reviewer in the discussion. However, authors also point out that theirs is the first algorithm which has instance wise O(log n) approximation to the best adaptive rate for arbitrary graphs.

 I believe this was an open problem. Previous works like [SMG+20] could not make a general argument due to their reliance on directed clique trees and some orientation properties of the directed clique trees. Current work takes a different approach using clique separators and authors very easily extend the results to interventions of bounded size (which was also not known in general).

3) Experiments were added in an anonymous drive folder (I would strongly suggest the authors to add a few to the main camera ready + put the rest in supplement and discuss in detail about runtime benefits etc. Currently the only discussion is in the readme file).

For all the three concerns, I feel authors have adequately addressed the concerns. This paper simplifies adaptive interventional design with many interesting observations and generalizations in addition to particularly novel contributions to the verification problem.

Hence, I am positive about this paper.

To the authors: Please do include the figures and discuss the experiments in the camera ready. Your anonymous folder contents must go into the paper (split between main paper and supplement) at the very least. Authors may think their theoretical contribution is the main point of the paper. However, experimentally seeing competitiveness to the baselines AND runtime benefits for various graph sizes is an important contribution. Unlike many other theory results, interventional complexity is unlike sample or computational complexity. Therefore, actual gains do matter (even multiplicative constants) and I do appreciate authors putting in the effort during rebuttal. It has definitely helped with one of the chief reviewer concerns.






**Award:**

No

---

### Decision · Program_Chairs · 2022-09-14

Accept